# Missing the (Tipping) Point: The Effect of Information about Climate Tipping Points on Public Risk Perceptions in Norway

Christina Nadeau[1], Manjana Milkoreit[2], Thomas Hylland Eriksen[3], Dag Olav Hessen[1]

[1]Centre for Biogeochemistry in the Anthropocene, University of Oslo, Oslo, 0316, Norway
[2]Department of Sociology and Human Geography, University of Oslo, Oslo, 0317, Norway
[3]Department of Social Anthropology, University of Oslo, 0317 Oslo, Norway

*Correspondence to*: Christina Nadeau (christina.nadeau@mn.uio.no)

**Abstract.** Climate tipping points are a topic of growing interest in climate research as well as a frequent communication tool in the media to warn of dangerous climate change. Despite indications that several climate tipping points may be triggered already within 1.5°C to 2°C warming above pre-industrial levels, there is limited research about public understanding of climate tipping points, the effects this knowledge (or lack thereof) may have on perceptions of risk related to climate change, and the corresponding effects on behaviour and public policy support. The emerging scholarship on learning, communication, and risk perceptions related to climate tipping points provides confounding evidence regarding the psychological and behavioural effects of information about climate tipping points. It remains unknown whether and under what conditions this knowledge increases concern, risk perceptions, and action intentions. In this study, we assess the current state of knowledge about climate tipping points among Norwegians using an online survey. We study the comparative effects of communicating about climate tipping points and climate change more generally on risk perceptions among participants with a survey-embedded experiment. Norway is an interesting case with its fossil-based economy and high level of education. We find that familiarity with climate tipping points is low among Norwegians; only 13% have good knowledge in the sense that they know an example or characteristic of climate tipping points. Information about tipping points had somewhat stronger, yet overall, very small, effects on participants' risk perceptions compared to general information about climate change, moderately increasing concern. We discuss our findings, and the implications, and suggest directions for further research.

## 1.0 Introduction

Efforts to mitigate climate change require urgent attention from both policymakers and the general public (IPCC, 2022). Despite recent progress, such as the acceleration of growth in renewable energy markets (IEA, 2022), global climate action continues to be insufficient to reach international objectives. While future warming projections have narrowed, pathways towards 1.5°C - 2°C futures do not appear credible without rapid, large-scale transformations of human systems (Kuramochi et al., 2022). Among the many reasons for this inadequate response to the climate challenge (Stoddard, 2021), public risk perceptions have played an important role. Public risk perceptions affect public support for climate policy and action (Bergquist et al., 2022; Drews & van den Bergh, 2016), and public support is a key condition for climate policy adoption, especially at the local scale (Yeganeh et al., 2020). When Lenton et al. published their seminal paper introducing the concept of climate

tipping elements in 2008, they argued that "society may be lulled into a false sense of security by smooth projections of global change" (p. 1792), i.e., that dominant conceptions of gradual, linear change might be at least partly to blame for the relatively low levels of concern about climate change and the persistent lack of urgency among publics and policymakers in the face of significant climate risks. This mental model of gradual change is now increasingly challenged by a growing body of scientific evidence for tipping points in the climate system. Here, we investigate whether exposure to knowledge on climate tipping points affects (i.e., increases) public risk perceptions of climate change in the national context of Norway.

Climate tipping points refer to non-linear change dynamics in large components of the Earth system. These so-called tipping elements can undergo state shifts in the sense that a change process that is initially gradual can reach a threshold (i.e., a tipping point), after which self-amplifying feedback mechanisms propel the system rapidly towards an alternative stable state. In many cases, these state shifts are irreversible on human timescales (Armstrong McKay et al., 2022; Lenton et al., 2008; Steffen et al., 2018). There is some evidence that multiple climate tipping points may be triggered within the temperature target range set by the Paris Agreement: 1.5°C to well below 2°C (Armstrong McKay et al., 2022; Schellnhuber et al., 2016; Wunderling et al., 2023). With recent projections indicating that global average temperatures could exceed 1.5°C in the 2030s (IPCC, 2021), perhaps even temporarily in this decade (WMO, 2023), climate tipping processes add new arguments for more ambitious climate action. This growing relevance has been reflected in more frequent appearances of climate tipping points in the assessment reports of the IPCC and in growing media coverage warning of dangerous climate change (Van der Hel et al., 2018).

However, it is yet unclear to what extent and how climate tipping points are understood by relevant audiences, how knowledge of climate tipping points affects climate risk perceptions, and whether and how this will influence behaviour or climate policymaking. The growing importance of climate tipping processes as a topic for climate risk communication and action is underexplored in research investigating public understanding, risk perceptions, and action orientations related to climate tipping points. Given the relative novelty of the concept of climate tipping points compared with the science of anthropogenic climate change, the level of public as well as policy maker knowledge is likely to differ between the two. There might also be significant learning challenges associated with climate tipping points (Renn, 2022), linked to the more general challenges of understanding complex systems. This context of uneven knowledge distribution and obstacles to learning has important implications for public risk perceptions and corresponding questions of behaviour change or political engagement. In contrast with now common studies of climate risk perceptions, existing knowledge and understanding of climate tipping points cannot be assumed.

Starting with the assumption that public knowledge of climate tipping points is likely less developed than more general knowledge of climate change, we investigate the current state of public understanding of this concept in Norway. Further, we study the effects of information about climate tipping points on climate risk perceptions compared with the effects of conventional climate change communication. We base our experiment in Norway, which is an interesting case by being a major producer of oil and gas, having a fossil-based economy, a high level of education, and yet, high levels of climate scepticism and inattention (PERITIA, 2022; YouGov, 2019). The following section (2) briefly reviews the literature on climate

risk perceptions, discussing whether and how climate tipping points present novel and specific challenges for this scholarship, and outline Norway as a case study for this research. Section 3 outlines our methodological approach, followed by a presentation of our results (4), discussion (5) and conclusion (6).

## 2.0 Climate Tipping Points: A Challenge for Climate Risk Perception Research

We briefly review the vast scholarship on climate risk perceptions, focusing on the role of knowledge and highlighting insights most pertinent to tipping points (2.1). In section 2.2, we describe the characteristics of climate tipping points that might affect public risk perceptions differently than climate change more generally. This is followed by a deep dive into the still limited literature on risk perceptions relating specifically to climate tipping points, where we identify hypotheses and existing, inconclusive evidence for the effects of exposure to information about climate tipping points on public concern about climate change (2.3). We conclude this section by bringing focus to our research location, Norway, and why climate risk perceptions in a petroleum-based economy provides an interesting case study for our research (2.4).

## 2.1 Climate Change Risk Perceptions

Climate change risk perceptions refer to individuals' subjective understandings, beliefs, and evaluations of the potential risks and impacts associated with climate change. It encompasses how people perceive the likelihood, severity, and personal relevance of climate change-related impacts. Perceptions of risk are subjective and influenced by several factors, such as personal experience, value orientation, emotion and affect, social norms, and knowledge (Salas Reyes et al., 2021; van der Linden, 2015). Given this complexity, it is important to consider how the multiple factors interact (Capstick & Pidgeon, 2014; Kahan et al., 2012).

Climate change presents a range of risk perception challenges, especially because it operates on long time horizons, and is perceived as a slow, incremental, and controllable phenomenon (Foz-Glassman, 2015; Sterman, 2011; Weber, 2006). Since many climate change impacts are expected to occur in the distant future, psychological distancing has played a prominent role in climate risk perception research (Jones et al., 2017; Spence et al., 2012). The psychological distance of climate change, e.g., in spatial or temporal terms, is often considered a barrier to climate action, although evidence for this claim has been inconsistent (Keller et al., 2022). More recent polling data and reviews suggest that the psychological distance of climate change might have been declining over the last few years; in many countries, the majority of polled citizens no longer perceive climate change as a distant threat (van Valkengoed et al., 2023). Nevertheless, climate change is associated with overall lower risk perceptions (Sterman, 2011; van Beek et al., 2022; Weber, 2006) than more abrupt and tangible phenomena, such as the Covid-19 pandemic (Manzanedo & Manning, 2020; Hochachka, 2020).

Knowledge plays a crucial role in climate change risk perceptions. Scientific knowledge - what is accepted as fact by the scientific community based on specific standards of knowledge production - can be distinguished from 'public knowledge' - that which people believe to be true and act upon. Here, we are interested in the latter. However, the phenomenon we investigate

occurs at the intersection of the two kinds of knowledge where the public communication of recent scientific insights (new
knowledge) is expected to create learning and belief revisions among the public.
In the context of risk perception research, van der Linden (2015) categorises knowledge as a cognitive factor, which differs
from experiential factors, socio-cultural influences, and demographics. Scientific knowledge of the risk source is the
foundation for understanding climate change, for identifying and evaluating related risks, and for counteracting misconceptions
(Majid et al., 2020). Some studies have shown clear correlations between instruction, information and knowledge on the one
hand and climate risk perception on the other (Aksit et al., 2018; Milfont, 2012; van der Linden, 2015; Xie et al., 2019), while
others caution that there is little evidence that knowledge is a strong indicator of pro-environmental behaviour (Kollmuss &
Agyeman, 2002) and that the relationship between knowledge and risk perceptions of climate change are more complicated.
Importantly, knowledge interacts with other variables that shape risk perceptions, especially with political belief and value
systems. Adherents to different political ideologies or cultural worldviews experience risks related to climate change very
differently (Kahan, 2012) driven by dynamics of motivated reasoning to protect a person's identity and core values. In Norway,
the high fossil-fuel dependency of the economy combined with a persistent governmental policy that the fossil fuel industry
should be developed, not liquidated (Redjeringen, 2022), no doubt plays a role. Individuals with a high degree of knowledge
of climate change can be found across the entire range of risk perception, from the alarmed to the dismissive (Capstick &
Pidgeon, 2014; Kahan et al., 2012). Norgaard (2006; 2011) argues that it is not a lack of information that reduces risk
perceptions of climate change, but a psychological need to keep threatening information at a distance, informing her theory on
socially organised denial.

## 2.2 Risk-relevant Characteristics of Climate Tipping Points

Modern science on anthropogenic climate change is over half a century old, whereas the term climate tipping points first
emerged less than two decades ago, in reference to Arctic ice sheet dynamics (Holland et al., 2006; Lindsay & Zhang, 2005;
Winton, 2006). Since then, the use of the term tipping point and corresponding body of knowledge in the climate sciences has
grown rapidly (Milkoreit et al., 2018). Different definitions of climate tipping points exist, and often identify a common set of
characteristics of climate tipping processes (Armstrong McKay et al., 2022; Milkoreit et al., 2018; van Beek et al., 2022), in
our study we focus on the following characteristics: multiple stable states, abruptness (non-linearity), self-amplifying (positive)
feedback mechanisms, and limited reversibility (or hysteresis). Some of these characteristics, especially as non-linearity and
limited reversibility, present significant aberrations from traditional conceptions of climate change as slow, incremental, and
controllable. As such, climate tipping points and impacts caused by tipping processes present complex uncertainty regarding
the potentially severe risks.
A **state shift** is the core characteristic of a tipping process, such as the potential transformation of the Amazon rainforest into
a grassland (Lenton et al., 2023). State shifts imply a reorganisation of the system in question, changing its main characteristics,
relationships between key entities, and functions. From a human perspective, this type of change process is fundamentally
different compared to incremental increases in temperature, sea-level rise, or even extreme events. System state shifts
permanently remove the current environmental conditions for human life and social organisation, likely forcing large-scale
social reorganisations as well.
**Non-linearity,** i.e., self-perpetuation and acceleration of change driven by **positive feedback mechanisms,** is a feature of
complex systems. Feedback mechanisms involve a closed loop of causality in which the change in a system is amplified
(mathematically positive) or dampened/balanced (mathematically negative) (Lenton et al., 2023). Tipping points occur where
positive feedback mechanisms overwhelm the balancing negative feedback mechanisms in a system, leading to self-
perpetuating and amplifying the initial change, forcing a rapid transition in a non-linear manner from one stable state to another
(Armstrong McKay et al., 2022; Lenton at al., 2023). Typically, humans tend to comprehend time and cause-and-effect
relationships in a linear manner (Dessai & van der Sluijs, 2007) and struggle to understand non-linear changes (Pereira &
Viola, 2018). While the climate system is complex, this linear model of causality has 'worked', given the well-established
linear relationship between the amount of greenhouse gases in the Earth's atmosphere and average global temperatures, which
is evident in IPCC scenarios (IPCC, 2022). However, the linear model cannot explain non-linear tipping dynamics, which
present distinct learning challenges (Plate, 2010; Milkoreit, 2015; Renn 2022). Related to the challenges of learning about
tipping risks is the observation that systemic risk perceptions are subject to attenuation and underestimation (Schweizer et al.,
2022). As Schweizer et al. note (2022, p. 1458) "they [systemic risks] are less easily understood and due to their complexity
and nonlinearity, less present in the mental representation of most people".
**Limited reversibility** implies that tipping processes and the changes they create cannot be 'undone' easily in the sense that
the system in question will not return to its initial state even if the driver of change is removed. For example, an ice sheet might
reach its tipping point and accelerate melting at a global temperature increase of 1.5C above pre-industrial levels. Even if
global temperatures were later reduced to below 1.5°C again, the ice sheet would not regain its mass. Reversing tipping
processes is possible, but requires different conditions (e.g., a return to much lower global temperature in the example of the
ice sheet), and, in many cases, is not achievable on timescales that are relevant for humans. Limited reversibility could have
significant and undesirable psychological and emotional effects, including the weakening of agency beliefs, the creation of
feelings of powerlessness, anxiety, fear, or dread (Milkoreit, 2014), leading to disengagement and avoidance (Norgaard, 2006;

154    2011).

In addition to these four, there are a number of additional features of tipping processes that might affect risk perceptions in a
predictable way. Like climate change more generally (Enserink et al., 2013; Marx et al., 2007), knowledge about climate
tipping points is subject to several types of **uncertainty**. Key uncertainties pertain to when (under what specific conditions)
different tipping points will be reached (Sterman, 2011), how long various state shift processes will take, and what kinds of
impacts they will have over time and in which places. Recent assessments conclude that some climate tipping points can be
triggered at +1–2°C of warming (Armstrong McKay et al., 2022). Given that global average temperatures could exceed 1.5°C
as soon as the 2030s (IPCC, 2021; WMO, 2023), the risk of triggering some climate tipping points may be "dangerously close"
(Lenton et al., 2019, p. 529).

While tipping processes are abrupt, they can occur over long timescales from a human perspective. These timescales differ for each tipping element, and the transition from one state to another can last from years (e.g. coral reefs) to decades (Amazon rainforest), or millennia (ice sheets), while the effects may last substantially longer (Armstrong McKay et al., 2022). Some of these timescales are short enough that human societies would struggle to adapt to the induced environmental pressures (Alley et al., 2003; Brovkin et al., 2021); however, they are of sufficient duration to invite psychological dynamics, like distancing (Spence 2012), and discounting (devaluing) of future impacts (Dasgupta, 2008).

The potential impacts of climate tipping points are underexplored in the scientific literature but knowledge about these could exert distinct effects on risk perceptions. There is general agreement that triggering climate tipping points will magnify well-established impacts of climate change (OECD, 2022), meeting the description of "dangerous climate change" (Armstrong McKay et al., 2022, p. 7). These risks include faster sea level rise, increased intensity of extreme weather events, and abrupt ecosystem shifts (Lenton et al., 2009; OECD, 2022; Wang et al., 2023), which could significantly affect human welfare, threaten global food and water security, and destabilise societies (OECD, 2022). Further, each tipping element has a certain potential to contribute to tipping cascades, which could destabilise multiple systems and ultimately have global reach (Kriegler et al., 2009; Lenton et al., 2019; Wunderling et al., 2021). These impact characteristics, especially negative impact amplification, could lead to a more negative assessment of the future, increasing concern among the public.

## 2.3 Perceptions of Climate Tipping Point Risk

The literature on risk perception and communication specifically related to climate tipping points is far more limited than the voluminous body of work on climate change more broadly. Initially, scholars expected climate tipping points to have significant effects on risk perceptions among the public and policy makers, likely increasing concern. For example, Russill and Nyssa (2009) suggested that communication related to climate tipping points could encourage audiences to include non-linearity in their mental models of climate change (i.e., the potential for rapid changes), and as a result reevaluate their risk perceptions. Nuttall (2012) argued that the looming threat of climate tipping points creates anticipation for the future, and that this heightened attention to long-term change can aid in guiding human action. Regardless of the psychological mechanism - worries about abrupt changes or lengthened time horizons - the hypothesis that climate tipping points would increase risk perceptions created hope that they might counter mitigation inertia (Gardiner, 2009) and boost climate action.

An early study by Lowe et al. (2006) provided some evidence for this hypothesis, finding that participants were more concerned about and willing to act on climate change after watching the 2004 film "The Day After Tomorrow", which depicted a fictional rapid cooling scenario due to changes in the Atlantic Ocean current. More recently, van Beek et al. (2022) investigated changes in risk perceptions related specifically to climate tipping points using a serious game. While their quantitative analysis did not show significant effects of the intervention (possibly due to ceiling effects), they argued based on a qualitative analysis that an increase in concern and perceived seriousness of climate tipping points could be observed, even among an audience with extensive climate change knowledge and a high baseline of concern - scientists and representatives of NGOs involved in climate change negotiations.

An additional hypothesis is emerging from the recent empirical work: information about climate tipping points might have no meaningful effects on public climate risk perceptions at all. The early work by Bellamy and Hulme (2011) already indicated that higher levels of concern were limited to a distinct social group sharing a particular worldview that is also associated with higher levels of concern for climate change. Then, a recent study by Formanski et al. (2022) investigated risk perceptions (and other beliefs) regarding climate tipping points with an experimental research design similar to ours. Focusing on one particular characteristic of tipping points, they studied whether participants who were given information about non-linear climate change processes would have qualitatively different risk perceptions than those presented with a common incremental change narrative. Formanski et al. (2022) found no difference between climate risk perceptions related to linear versus non-linear portrayals of climate change (based on a short message combined with a graphical depiction of future temperature change).

Each of these hypotheses - increased concern and no effects - would have different implications for public communication related to climate tipping points. Given the limited and mixed evidence for risk perception effects so far, science communication and media reporting on climate tipping points lacks guidance.

Here, we seek to advance empirical understanding of this phenomenon, pursuing in particular questions about the role of knowledge as a foundation for climate risk perceptions. A number of prior studies have indicated limited public and policy maker awareness of the concept of climate tipping points. For example, Milkoreit (2019) reported limited knowledge among climate negotiators in 2018, and Bellamy's survey of the UK public (2023) showed that more than a quarter of respondents were unfamiliar with climate tipping points in 2022 despite increased media coverage of the topic. At the same time, systemic risk scholars have argued that tipping points present specific learning challenges and tend to receive less public attention than they merit (Schweizer et al., 2022; Renn, 2022). Hence, understanding the state of public knowledge, limitations in understanding, and misconceptions is important to support future communication efforts related to climate tipping points.

## 2.4 The Norwegian Context

The focus of this study is on climate change risk perceptions in Norway, a small, oil-rich nation that perceives itself as a genuinely concerned nation about climate change (Painter, 2013; Eckersley 2016). Norway's state-owned company Equinor is engaged in oil and gas extraction primarily for export purposes (Griffin & Heede, 2017), making Norway a significant contributor to anthropogenic GHG emissions. The Global Footprint Network (2023) reported that Norway also had one of the highest carbon footprints per capita in Europe. Contrastingly, Norway is often cited as an example of reaching a consumer tipping point in the purchase of electric vehicles, pointing to the country as a leader in decarbonising their transport system (IEA, 2019; Sharpe & Lenton, 2021).

Recent polling data suggest that Norway is home to a significant amount of climate scepticism, with around 24% of Norwegians not believing in anthropogenic climate change (Krange et al., 2019; YouGov, 2019). At the same time, Norway is facing visible signs of climate change, with increased rainfall and frequency of landslides along the West Coast (Hanssen-Bauer et al., 2015). However, research conducted by KANTAR (2020) found that only a third of the population in Norway

noticed the ongoing consequences of climate change around them. Along with current and future effects of climate change, Norway is likely to be physically affected by the impacts of a number of identified climate tipping points, such as thawing of mountain glaciers and permafrost, shifting boreal forests, melting ice sheets, and ocean circulation destabilisation. Rapidly declining glaciers is likely not perceived as a real risk, and the same holds for the vanishing permafrost in northern parts of the county (e.g. Finnmark) or on the island of Svalbard, since it does not pose critical risk to human infrastructure.

Public knowledge of climate tipping points would have to be based on the consumption of media reporting. As Bellamy shows (2023), media coverage of climate tipping points has significantly increased in international English language reporting over the last twenty years, especially since 2018. To understand whether and to what extent Norwegian newspapers have been covering the topic of climate tipping points relative to general climate change, we conducted a quantitative analysis of Norwegian media using the database available through the National Library of Norway. Our search covers the time period from 2005 to 2022 and over 100 Norwegian press newspapers (local and national) for articles containing the following terms in Norwegian: global warming, climate change, and climate crisis, tipping point, and climate+tipping point ("global oppvarming", "klimaendring", "klimakrise", "vippepunkt", "klima+vippepunkt"). As expected, we found substantially more media content on climate-related terms without mentions of tipping points. The first article mentioning climate tipping points was published in April 2006 in the newspaper Klassekampen and focused on irreversible climate changes. It was entitled "Is it too late to turn back?". Coverage of the subject remained limited (less than 50 articles per year) until 2017, and has been expanding since 2018, mirroring Bellamy's analysis of British and international news media.

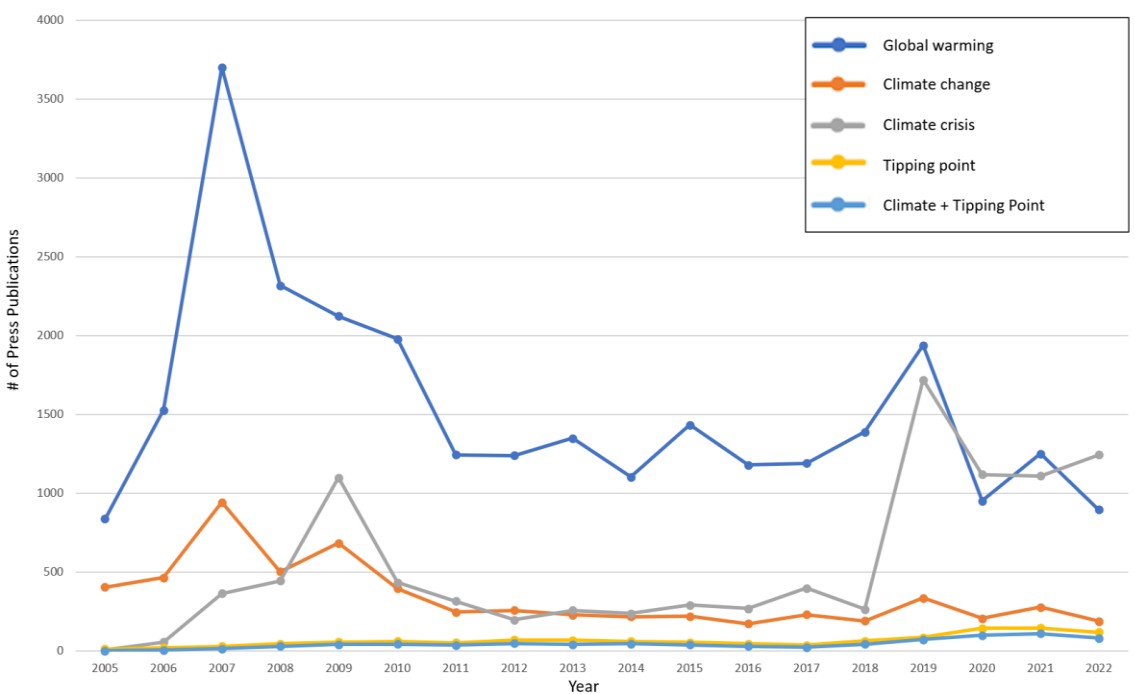

**Figure 1: Norwegian mentions in press newspaper articles of climate change and climate tipping points.**

Mentions of climate tipping points likely became more common after the publication of the Norwegian popular science book "The World on a Tipping Point" (Verden på vippepunktet) by Hessen (2020), and which received wide public attention. Based on these findings we expect knowledge of climate tipping points in the general population to be lower than general climate change.

**3.0 Research Design and Methods**

**3.1 Survey Design**

Our study sought to answer the following research questions:

1. What is the level of knowledge of climate tipping points among Norwegians?
2. To what extent does the information on climate tipping points increase concern about climate change?

To answer both questions, we conducted a web-based survey with an embedded experiment, which was implemented by a third-party polling service in Norway. The survey consisted of three parts. In part 1a, all participants were asked a series of questions about their climate change risk perceptions, including concern, seriousness of climate change and the need to act (see Appendix B, q1r1, q1r2, and q1r3) using a 5-Point Likert scale, ranging from 1 "completely agree" to 5 "completely

disagree" providing a moderate level of granularity for respondents to express their opinions. The internal consistency reliability of the questions measuring climate change risk perceptions (CCRP) was assessed using Cronbach's alpha. The calculated value obtained for Cronbach's alpha was $\alpha = 0.897$ indicating strong internal consistency reliability among the questions measuring CCRP, suggesting that the questions are highly correlated and likely measure the same underlying construct effectively. Part 1b assessed knowledge about climate tipping points (research question 1). Our survey design sought to gauge general familiarity with the concept without providing or asking for a definition or description of characteristics of the phenomenon. The latter is not necessarily a reliable indicator of public understanding of a difficult scientific concept, where multiple definitions exist within and across disciplines (Milkoreit et al. 2017) and continue to be debated. Relying on free recall and knowledge self-assessment, we used a Likert scale to assess perceived familiarity with the concept of climate tipping points ("vippepunkter"). The scale items ranged from "never heard of it" to "know it well", including 'I am not sure/I do not know". If the participant indicated at least some familiarity with climate tipping points ("know it well", "a little familiar", "neutral"), they were asked if they could give an example (yes/no question). Those who answered yes were prompted to provide a written example to demonstrate actual knowledge (see Appendix B, q3b). This approach was grounded in cognitive theories of concepts and learning (prototype and exemplar theories), which rely on category formation based on examples of the phenomenon in question (e.g., Hampton 2006; Park 2013). We assumed that providing an example was an easier cognitive task than providing a definition. This design, presenting increasingly challenging questions that combine self-evaluation, recall and a knowledge 'check', allowed participants to reveal their degree of familiarity with the concept.

This design contrasts with Bellamy's (2023), who presented survey participants with a definition and ten examples of climate tipping points identified in the literature and asked for self-reported familiarity with these. We purposefully did not present participants with a definition or examples but sought to elicit information about their knowledge based on participants' ability to recall examples themselves. This limits the influence of biases like socially desirable responding (e.g., projecting knowledgeability).

For part 2 of the survey, participants were randomly assigned to one of two experimental conditions, receiving different kinds of climate change information in text form. The participants in Group 1 were asked to read a text that introduced them to the concept of climate tipping points and included specific characteristics of tipping points identified in the literature (non-linear and abrupt change, irreversibility of climate change, system interactions and domino effects). The text for Group 2 presented more general information about climate change without terminology pertaining to climate tipping points. The texts were comparable in length - each took 2-3 minutes to read - and with the same intended linguistic style and difficulty. They were significantly longer than the texts used by Formanski et al., (2022), but still short relative to a common news article. The texts were presented in Norwegian (English translations in Appendix A).

In part 3, all participants were asked the same questions presented in part 1a about their risk perceptions related to climate change (see Appendix B, q5r1, q5r2, and q5r3).

The responses to the survey were managed using SPSS data files, and later converted into Microsoft Excel spreadsheets for processing and analysis in R. Our results were verified by an unbiassed and impartial third party.

The survey design is visualised in Figure 2.

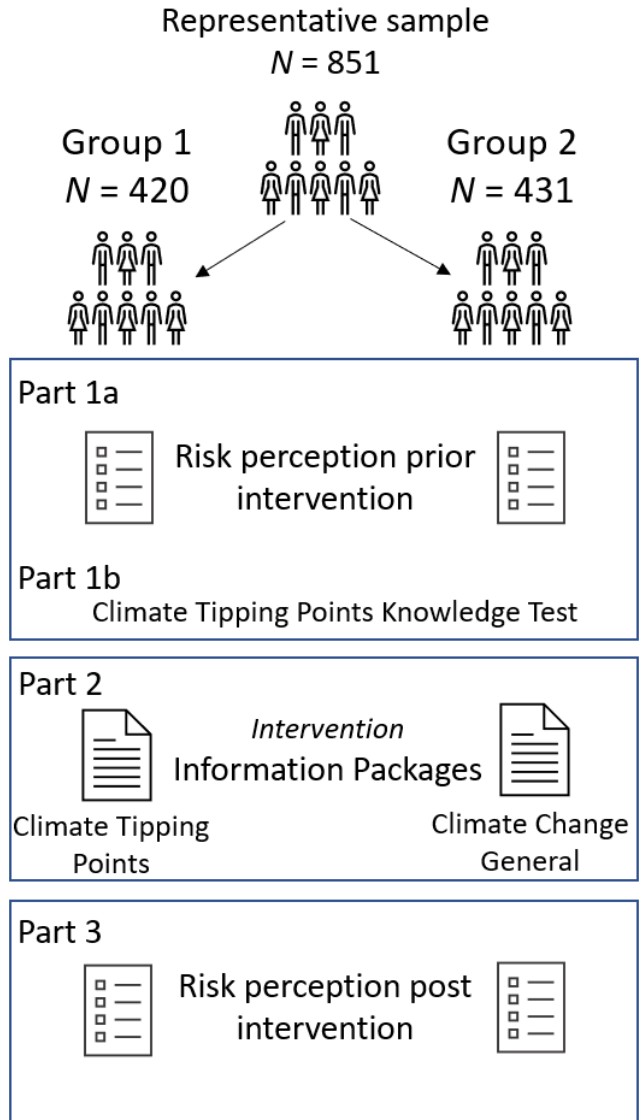

**Figure 2: Survey Design Outline.**

Our survey was conducted by an external Norwegian data collection unit (Opinion) in October-November 2022. A quota
sample of participants was recruited from a pool of over 8000. Our sample included 851 adults ranging from 18 – 91 years of
age with a 50/50 split between men and women from all regions of Norway (northern Norway, 9%; central Norway, 14%;
west Norway, 20%; east Norway, 30%; south Norway, 14%; Oslo, 13%) and did not favour any specific characteristics (pro-
environmental views, political orientation, level of education etc.). It is important to note, that we attained a "quota" sample
and not necessarily "nationally representative" sample of the Norwegian population which is limited mainly to gender, age,
and geographic location. This leaves space for some members of the population to be excluded, such as immigrants,
international students, refugees, people with disability, and non-binary participants. This limits our study in reflecting certain
dimensions of diversity and should be considered in future recruitment processes. While our study provides valuable insights
within the scope of our sample, we acknowledge the need for caution when generalising the findings beyond our specific
sample group.
**3.2 Analysis**
We used a primarily qualitative approach to assess knowledge and a statistical analysis to analyse changes in risk perceptions.
● Knowledge
Using data from pre-intervention questions about knowledge, and adjusting Milkoreit's (2019) approach, we categorised
participants into four different levels of knowledge (no knowledge, incorrect knowledge, some knowledge, good knowledge).
Participants who indicated that they were not familiar with the concept of climate tipping points at all, or that they had 'little
knowledge' were categorised as having "no knowledge" (category 1). All remaining participants were considered to have some
knowledge based on their self-assessment. Those who indicated that they could not provide an example were categorised as
"some knowledge, but not demonstrated" (category 2a). Similarly, if a participant answered yes (indicating a self-assessed
ability to offer an example) but then did not provide a response to the prompt for an example, they were also categorised as
"some knowledge, but not demonstrated" (category 2a). It is possible that some of these participants had some form of
knowledge about climate tipping points and their characteristics but did not provide written examples for reasons other than
inability to recall this information, e.g., time constraints or a general unwillingness to answer open-ended questions. It is also
possible that they would have been able to demonstrate knowledge if we had specifically asked for a definition, a description
or other associated ideas. However, we assumed that the most likely reason for not providing any text was the inability to recall
and provide relevant information.
Participants who responded to the prompt to provide an example of a climate tipping point were categorised based on the
content of their answer. The answers contained both correct and incorrect examples of tipping elements, such as "arctic sea
ice" or "Gulfstream", and more general (unelicited) descriptions of the concept, such as "point of no return" or "an irreversible
event". These open-ended answers were coded distinguishing "some but incorrect knowledge" (category 2b) and "good
knowledge" (category 3) (see codebook in Appendix C). Individuals who had provided either a false example or a description
that did not contain any tipping-point characteristics identified in the literature were placed in category 2b; those who had
identified at least one correct example of tipping points or correctly described one or more characteristics of tipping points
were placed in category 3.
1. No knowledge: self-reported lack or limitation of familiarity with climate tipping points.

2a. Some knowledge, but not demonstrated: self-reported familiarity with climate tipping points but answered NO to the question about ability to provide an example OR answered YES to question about ability to provide an example but did not respond to the prompt to provide an example.

2b. Some knowledge, but incorrect: self-reported familiarity and ability to provide an example, but inability to provide a correct example of a tipping point or any description (feature) that could be associated with climate tipping points.

3. Good Knowledge: identified one or more correct examples or at least one correct feature of climate tipping processes.

These categories are based on rather moderate expectations of what constitutes knowledge and maximise assumptions in favour of knowledge. This approach likely overestimates the state of actively usable knowledge that could shape risk perceptions. Especially category 2 likely includes participants with a very limited understanding of the phenomenon.

While none of our survey questions asked participants to identify characteristics of climate tipping points, we found it meaningful to code these open responses in addition to specific examples of climate tipping points as instances of demonstrated knowledge. We also coded these unexpected responses for common characteristics of climate tipping points - alternative stable states, threshold, positive feedback, non-linearity, limited reversibility (Milkoreit et al., 2017) - and analysed how frequently each of these were mentioned (see Appendix D and Table D.1).

We assessed public familiarity with specific tipping processes by counting how often they were mentioned as examples by participants. Here it was necessary to distinguish types/classes of tipping elements and specific examples within each type. The different types included cryosphere tipping elements/ice sheets, circulation patterns in the oceans and atmosphere, and biosphere tipping elements. Some participants referred to these types of tipping elements, while others provided more specific examples, such as the West Antarctic Ice Sheet or the Amazon rainforest. Based on these counts, we assessed which known tipping elements the public is currently most familiar with.

● Risk Perceptions

Our survey data were quantitatively analysed using data analysis tools in Microsoft Excel and R in order to identify any effect on climate risk perceptions post-intervention between the two groups (analysis for covariance (ANCOVA analysis) and *t*-Test: two sample assuming unequal variances) and within the same group (*t*-Test: paired two sample for means). Significance tests were performed on the data in order to identify any statistically significant differences in responses on concern levels for climate change post-intervention.

**4.0 Results**

**4.1 General State of Knowledge of climate tipping points**

When asked about their self-reported level of familiarity with climate tipping points, 6% (n=53) indicated good levels of familiarity, 22% (n=183) indicated some familiarity, 15% (n=126) were neutral, 23% (n=196) indicated very little knowledge,

29% (n=244) indicated they had never heard of the term. 5% (n=49) answered they were unsure or did not know. Based on
our categorization, 52% (n=440) had no knowledge about climate tipping points. About 42% (n=362) of the participants
indicated some knowledge about climate tipping points and were asked whether they could give an example. More than half
of these (n=201) answered no. Among those who answered yes to this question, 8 did not provide a response. Hence, 25% of
participants (n=209) fall into category 2a - some but no demonstrated knowledge. This left us with 153 responses to the open-
ended question. When investigating the demographics of this group of respondents we found that the age group under 30 were
under-represented (12.4% n=19), while the age group 60+ were overrepresented (35% n=54). Additionally, we found that
respondents identifying as men were over-represented (64.7% n=99) while respondents identifying as women were under-
represented (35.3% n=54). The small number of participants (n=44, 5%) who provided incorrect responses, which included
false examples or descriptions of climate change generally, such as "global warming" or "increasing average global
temperatures", fall into category 2b - incorrect knowledge. Finally, 13% (n=109) demonstrated good knowledge, i.e., were
able to provide a correct example or identify at least one correct feature of tipping processes (see Figure 3).

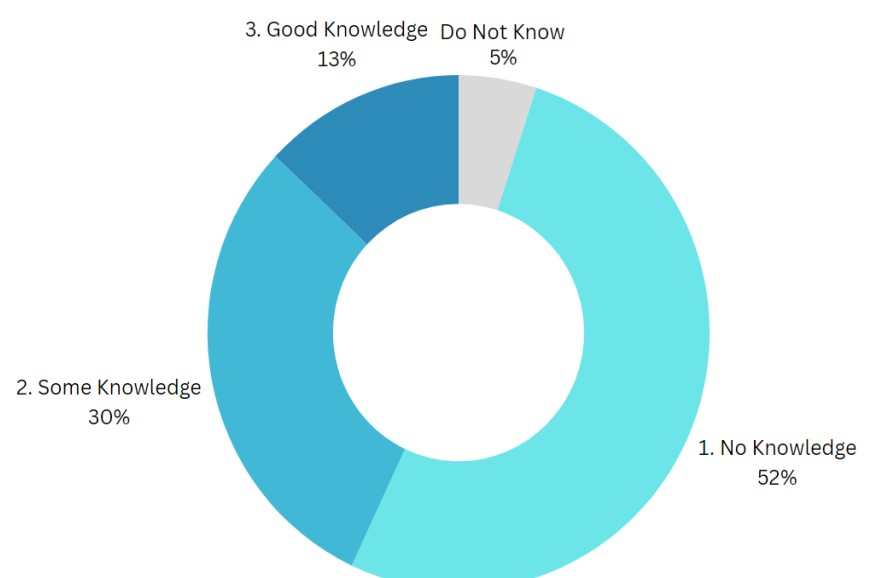

**Figure 3: Results on the level of knowledge of climate tipping points amongst participants.**

Given the self-reported lack of familiarity among 52% of the population (category 1) and the lacking ability to recall (correct) information
about climate tipping points among another 30% of participants (categories 2a and 2b), we argue that 4 out of 5 of Norwegians lack
knowledge of the concept in the sense that their understanding is insufficient to meaningfully inform a person's risk perceptions related to
climate change (see Appendix C for more details).

After the intervention participants were asked whether the text, they had read contained information that was new to them (see Appendix B q4r1). A higher percentage (27%) of participants who read the text on climate tipping points agreed that the information was new to them compared with the general climate change group (17%). This difference was highly statistically significant (t (848) = -5,98, p < 0.05). However, this result does not align with the result of participants indicating a lack of familiarity with the concept of climate tipping points in part 1b of the survey (more than 50%). This disparity between initial self-report of knowledge and post-experimental assessment of the information's novelty indicates reliability problems regarding self-report data, possibly linked to a desire for socially desirable responses. This difference could also be explained by the fact that the free recall of memorised information is a more challenging cognitive task than the recognition of previously encountered information. In other words, participants might not have been able to recall the definition or examples of climate tipping points in part 1b of the survey, but later remembered having heard or read about the concept when they encountered the materials provided for the experiment.

**4.2 Characteristics of Climate Tipping Points**

Our survey did not include an explicit question about the characteristics of climate tipping points, so we cannot draw any conclusions from our analysis about familiarity with characteristics in the Norwegian population. However, many of the participants, who were presented with and chose to respond to the prompt to provide an example of a tipping process n=153, 18%), responded by providing descriptive comments instead of or in addition to an example. These comments identified characteristics of climate tipping points and provided an unexpected opportunity to add a layer of analysis about current knowledge patterns within the more knowledgeable population segment. We calculated the frequency with which specific characteristics of tipping points were mentioned by participants in this sub-group of 153 participants to identify the most and least common features in public understanding (see also Table D.1 Appendix D).

Participants mentioned limited reversibility most frequently (n=46, 30% among this sub-group), with some using the term "irreversible" directly, or phrases such as "unable to turn back" or "point of no return". Feedback dynamics were the second most common characteristic mentioned (n=18, 12%) with phrases such as "self-reinforcing loops" or, more frequently, detailed descriptions of feedback loops, such as "less ice allows more light absorption which leads to more ice melting." Thirteen participants (8%) used terms including "threshold," "boundary" or "limit" that is crossed to refer to critical thresholds. Very few individuals mentioned non-linearity (n=8) or multiple stable states (n=4). For abruptness and non-linearity, participants used terms such as "escalating" to describe change or stated that climate change will happen "even faster." The idea of multiple stable states was described with the terms "unstable" "fluctuating," or "change from one system to another". Other features such as severe impacts and uncertainty were not mentioned often enough to be considered part of a common understanding.

**4.3 Examples of Climate Tipping Points**

By far, the most frequently identified type of tipping points were those related to the cryosphere - 71 mentions (provided by 61 participants, some mentioning multiple elements) referred to ice loss, especially the Greenland Ice Sheet and the Arctic Sea

Ice. The majority of these referred to "ice melting" or "polar ice" and "glaciers disappearing" without specific geographical
reference. Some participants referred to "glaciers" but did not specify if these were mountain glaciers specifically, therefore
these responses were coded as ice loss generally. More specific examples included "permafrost", the Greenland ice sheet, and
the loss of sea ice in the Arctic.
The second most frequent type of tipping element was circulation patterns (7 mentions) followed by biosphere components (4
mentions).  Mentions of circulation patterns included mentions of "the Gulf stream" or "ocean currents", and one mention of
"air currents". Regarding biosphere components, only one person identified the "coral reefs" and two the "Amazon rainforest".
The results are summarised in Figure 4 (data used for this figure can be found in Appendix C table C.3).

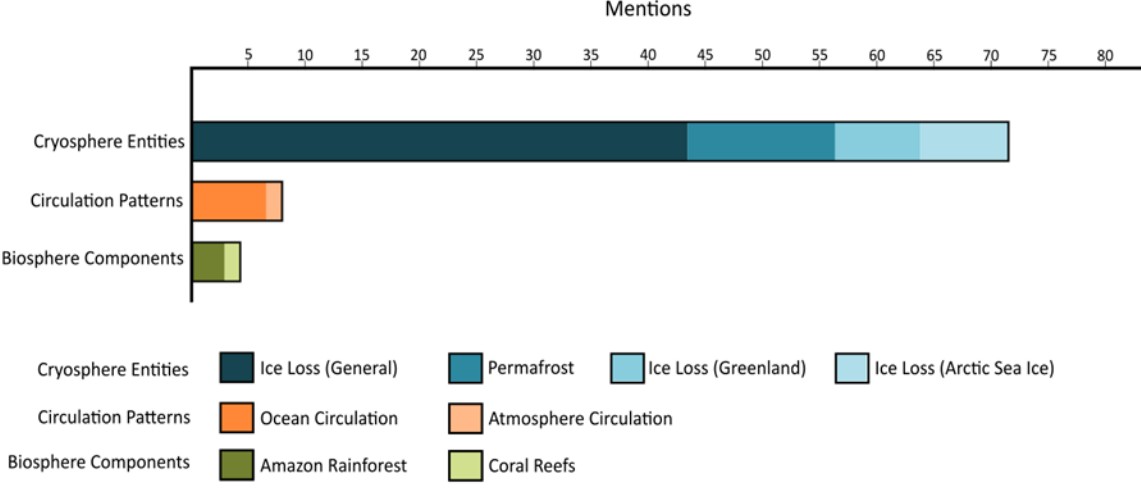

**Figure 4: Most commonly and correctly identified climate tipping elements by participants.**

### 4.4 Effect of Information about Climate Tipping Points on Level of Concern for Climate Change

All participants were asked about their concerns about climate change before and after our intervention (survey questions q1r1
and q5r2 found in Appendix B). A series of statistical analyses were performed in R. Firstly an ANOVA comparing the post-
intervention scores to the pre-intervention scores which shows a statistical significance ($F$ (1, 848) = 1962.1, $p < 0.000$). A
second ANOVA test found there was also a statistical significance when comparing Groups 1 and 2 post-intervention ($F$ (1,
848) = 3.998, $p < 0.05$ [= 0.046]), but not between Groups 1 & 2 pre-intervention ($F$ (1, 848) = 0.892, $p = 0.345$). This indicates
that our experimental treatment (reading a brief text with information about climate tipping points) significantly contributed
to a change on risk perceptions of climate change in Group 1, while it (text on climate change) did not have an effect on Group
2. An ANCOVA combining pre-intervention and post-intervention comparisons, as well as group comparisons resulted in
statistically significant differences in both treatment differences (pre/post) ($t = 44.301$, $p < 0.000$) and group differences ($t = -$
2.201, *p* = 0.028).   For data used in our ANCOVA analysis, including mean values and standard deviations for each
experimental condition, see Table 1.

Table 1. Results of Statistical Analysis

|  | Group A Pre-Intervention | Group A Post-Intervention | Group B Pre-Intervention | Group B Post-Intervention |
|---|---|---|---|---|
| Mean | 2.46 | 2.37 | 2.54 | 2.54 |
| ± *SD* | 1.23 | 1.29 | 1.20 | 1.31 |


From our analysis, the effect size of the post-pre-intervention score difference without any adjustment for pre-scores is Cohen's
d 0.14 which would be considered very small according to Cohen (1988) with a confidence interval of 0.00 - 0.27. Furthermore,
we find the effect size difference between Group A and Group B in the standardised post-intervention score to be 0.08 standard
deviation units, based on Field (2013). The interpretation of our data finds that, while there is a statistical difference between
the two groups post-intervention, the effect is very small.
The difference in responses post-intervention for the two groups is illustrated in Figure 6. The biggest change in responses
before and after our intervention was that some who agreed before the intervention that they were concerned about climate
change, completely agreed that they are personally concerned after the intervention. Both Groups 1 and 2 saw shifts of this
nature, however, Group 1 who were presented with information on climate tipping points saw a higher degree of difference
post-intervention.

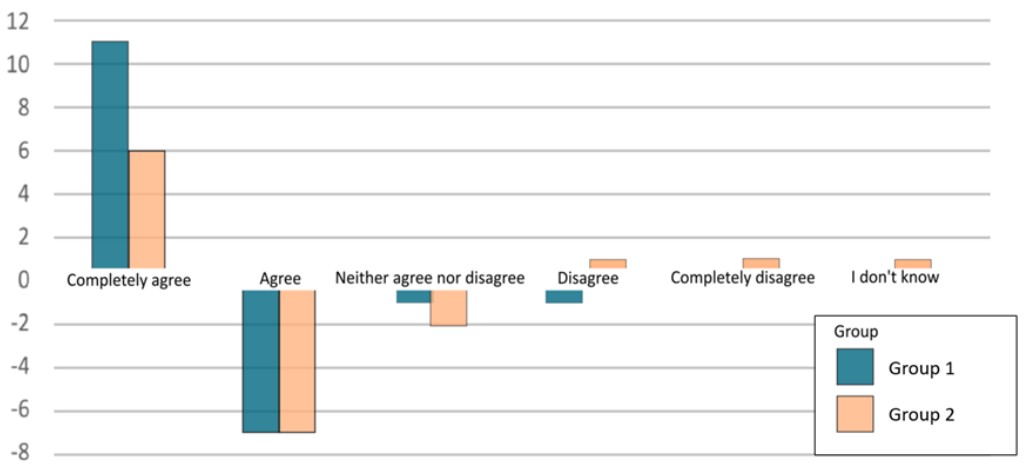

**Figure 5: Percentage difference in climate risk perceptions post-intervention per Group (Group 1: climate tipping points text; Group 2: climate change general text)**

A paired *t*-test for means was performed on Group 1 between their level of concern before and after the intervention in Microsoft Excel. The results indicate that the responses from Group 1 were significantly different post-intervention ($t$ (419) = 2.72, $p < 0.05$). The same test was carried out for Group 2, and it was found that the difference in response post-intervention was not statistically significant ($t$ (430) = -0.07, $p > 0.05$).

**5.0 Discussion**

Despite our expectations that knowledge of climate tipping points would be more limited than knowledge of climate change in general, we were surprised to find that more than 80% of respondents were not sufficiently familiar with the concept to provide an example. Ultimately, only 13% of Norwegians have an understanding of climate tipping points that might be sufficient to serve as a foundation for risk assessments and potential behavioural changes. Even within this more knowledgeable segment of the population, understanding of climate tipping points and familiarity with examples is heavily skewed towards a specific type of tipping process (ice loss). These findings indicate a potential knowledge gap between scientists and the general public with important implications for understanding risk perceptions, policy support, or behavioural change related to climate change. For the large majority of Norwegians, knowledge of climate tipping points likely does not yet affect judgements of climate risk.

Comparing our assessment to that of Bellamy (2023), the state of public knowledge in Norway appears to be significantly weaker than that in the UK. Bellamy reported that 25% of British study participants had not heard of any of the ten explicitly named climate tipping points before taking their survey, and that awareness of the issue is still low in the UK. These observations could be indicative of more limited media communication on climate tipping points in Norway compared to British and international press, but they could also be the result of different methodological approaches. Our research design relied on participants' free recall of examples while Bellamy provided survey participants with a list of ten tipping points and asked whether they had heard of these before. The latter is a less demanding cognitive task than open recall, which might account for some of the difference and suggest that familiarity in Norway might be higher than our findings reflect.

At the same time, Bellamy's findings strongly mirror our own regarding awareness of specific tipping elements. In both countries, there is significant variation, with highest scores for tipping elements in the cryosphere (e.g., over 50% of British participants were familiar with Arctic Sea ice loss) and the potential dieback of the Amazon rainforest, and lowest scores (under 20% in the UK) for the risk of AMOC collapse and boreal forest dieback. Cryosphere elements are the most commonly mentioned examples of climate tipping points in our study. This may be due to the fact that Arctic Sea ice was the first Earth system component to be associated with tipping points (Winton, 2006), and likely also its significance to Norway being proximal to the Arctic. The prominence of ice-related examples may also be due to the rather simple cause-and-effect relationship between higher temperatures and melting ice sheets, and the prevalence of cryosphere change in visual media reporting, e.g., eye-catching photos of polar bears on (disappearing) icebergs. The reasons for the differential popularity, recognizability of and attention to various tipping elements should be explored in future research. While Arctic summer sea ice is no longer considered to have a tipping point (Armstrong McKay et al., 2022), other elements of the cryosphere (e.g., Greenland Ice Sheet, West Antarctic Ice Sheet) remain policy relevant with significant impacts on human systems. More surprising is the lack of public awareness of the AMOC as a potential tipping point with potentially dramatic consequences for Norway and all Atlantic states. Recent studies find that the circulation system is at its weakest in 1600 years (Boers, 2021; Thornalley et al., 2018) and some has argued that it could cross a tipping point this century (Ditlevsen & Ditlevsen, 2023), yet this is a highly disputed worst-case scenario.

More generally, the shared finding that there is low public awareness of climate tipping points in these two countries indicates the learning challenges related to tipping processes as complex systems dynamics that defy mechanistic causal thinking. Media reporting on climate tipping points has been increasing over the last five years, but with limited effects on public understanding so far. Given this baseline of limited knowledge paired with cognitive and emotional barriers to learning, it is likely that our experimental intervention - a short, fact-based description of climate tipping points - had very small effects on risk perceptions because of its limited potential to contribute to learning and understanding.

Our analysis provides modest evidence for the hypothesis that climate tipping point communication can increase public concern about climate change compared to more conventional, linear descriptions of climate change (Russill, 2015). We observed that the strongest change in risk perceptions occurs among those who already are concerned about climate change, which aligns with findings by van Beek et al. (2022), although our survey-embedded experiment was significantly less

engaging than the serious game deployed in their study. Our results differ from recent findings by Formanski et al. (2022) who found no difference in risk perceptions between participants presented with portrayals of linear versus non-linear climate change. This difference may simply be due to our larger sample size (n=851 versus n=360), as small effects may become more significant when the sample size is larger. Formanski et al (2022) found that there may be no effect of non-linear portrayals of climate change on public risk perceptions, but if there is an effect it is likely to be very small. Our research finds that an effect may indeed exist, and was statistically significant in our sample, but the effect was indeed very small. We cannot claim that our results contrast with previous studies, but only that they differ and may indicate an effect may exist. One explanation for this difference might be that Formanski et al. focused on a specific characteristic of tipping points (non-linearity), which might not be the feature that generates most concern. Although we did not explicitly ask participants about their familiarity with different characteristics of tipping processes, and cannot draw any general conclusions from our data, limited reversibility was the most commonly identified feature of climate tipping points in the open responses provided by a subset of our study participants. This is similar to findings by Milkoreit (2019) from surveys with international policymakers, who also indicated most concern with irreversibility. It could be argued that people are more concerned about the permanence of losses rather than the speed of change, especially when limited reversibility is combined with the possibility of severe harm ('catastrophic risks'). While we cannot conclude this from our study, it opens up pathways for future research.

Major questions remain regarding how to best communicate the risks of climate change, balancing information about threats with motivation to act and managing a complicated medley of emotions, including fear and apathy. This discussion is particularly relevant for Norway with the paradoxical gap between the political acceptance of climate risks and continued oil exploration, as well the apparent widespread ignorance or denial of climate change (PERITIA, 2022; YouGov, 2019). While communicating risk based scientific predictions of potential tipping points should motivate climate action among both politicians and voters, the concept is hardly known, and partially misunderstood. One could argue that in Norway, a general feeling of safety, trust in government and technological solutions are widespread, creating a kind of hubris with regard to climate risks that can only be overcome by personal experience as the situation worsens (cf. Lujala et al., 2015). However, there is major potential to increase the scale and effectiveness of public communication about the risks of climate tipping points. Future research should seek to support the development of effective communication strategies, considering national differences, including differences in cultural worldviews (Bellamy, 2023).

**6.0 Conclusion**

Understanding climate change risk perceptions is crucial for effective communication, policymaking, and public engagement. Climate tipping points, while presenting a range of threats to societies, might also provide new communication tools and opportunities to reshape existing climate change narratives, public risk perceptions, engagement, and support for climate action.

Our study investigated the level of knowledge of climate tipping points among participants in Norway, and assessed whether information pertaining to climate tipping points has a different impact on climate change risk perceptions compared with information about climate change more generally. Among our findings, two stand out. First, we found a widespread lack of knowledge about climate tipping points among Norwegians, suggesting that the topic remains "new" for the majority of the population despite its increasing presence in the media. This situation merits further investigation, focusing on the question how to best support public learning and meaning making related to tipping points, including active learning strategies (Beek et al. 2022; Formanski et al. 2022) in the face of significant obstacles to informal learning.

Second, our results indicate a very small effect of information about climate tipping points on risk perceptions of Norwegians, while more general climate change information had no significant effect. We suspect that these limited effects are linked to the general state of public knowledge in Norway and the limited effects of our intervention on participants' understanding of the concept and its potential implications for human wellbeing. In other words, the reading materials provided to participants did - in most cases - not enable learning about tipping points in a way that affected existing risk perceptions. Our experimental treatment might have been ineffective as a learning device. Given the findings and limitations of our study, we recommend further investigation into how laypeople and decision-makers learn about the risks posed by climate tipping processes, and how (or if) knowledge about tipping points changes existing perceptions of climate change risk. Future work should explore in particular whether different modes of communication, engagement and learning have different impacts, e.g., active learning strategies such as serious gaming, passive learning through reading news or story-based information. This work might require more challenging experimental designs (e.g., game or storytelling workshops) coupled with in-depth interviews, focus groups, or observations.

Given that climate risk perceptions are shaped by multiple factors in addition to knowledge (Kahan et al., 2012; Libarkin et al., 2018), future research should also explore how (changes in) knowledge and understanding interact with other variables over time to investigate the complex psychological processes that may be triggered by forewarnings of climate tipping points. Future work should not only consider the role of cultural cognition in the adoption of this concept, but also emotional and social barriers to learning, such as psychological distancing, identity protection and socially organised denial (Norgard, 2011). Future research should also explore the relationship between climate risk perceptions and action gap.

**Funding Sources**

This work was supported by the UiO: Norden initiative at the University of Oslo and the Centre of Biogeochemistry in the Anthropocene (CBA).

**Competing Interests**

The contact author has declared that none of the authors has any competing interests.

## Acknowledgments

We would like to thank Opinion for their collaboration in this research and their role in conducting a public opinion poll in Norway.

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
