# Peer review of "Missing the (Tipping) Point: The Effect of Information about Climate"

_Earth System Dynamics, 2023_

## Author Response (AR1)

Response to Reviewers

We would like to thank the editor and all the reviewers for their engagement with our manuscript; we greatly appreciate their constructive feedback. We have significantly revised the manuscript to address all the issues that were raised. Below we provide detailed responses to the reviewers' comments and describe the changes we have made in the manuscript. Our line numbers reflect the location of the change in the clean manuscript.

**General & Technical Edits**

We have slightly edited the title of the manuscript, replacing "role" with "effect of information about" climate tipping points, as this more accurately describes the phenomenon we investigate.

**Reviewer 1**

**General comments**

*"[...] it is difficult to fully assess the quality of the manuscript and the underlying research given that critical information on the experimental treatment, questionnaire, codebook, and the data is missing."*

Response: The omission of this material in our initial submission was an oversight. We have included this important information in supplementary materials to the revised manuscript, such as the texts used as experimental interventions in our study (translated from Norwegian) in Appendix A, details of the survey questions (translated from Norwegian) in Appendix B, our codebook, and the responses from the open answer questions (translated from Norwegian) in Appendix C, and a supplementary figure from our ANCOVA analysis in Appendix D.

*"[...] Some sections of the manuscript seem half-finished (e.g., some headers and sentences appear twice)."*

Response: We have corrected the repetitions where headers appear twice and other such errors throughout the manuscript.

**Introduction**

*"Lines 28-30: "Among the many reasons for his [sic!] inadequate response to the climate challenge (Stoddard 2021), public risk perceptions and the corresponding support for climate action have been paramount." First of all, I would not use the term 'paramount' – other factors can be considered as equally important as policy support and risk perceptions.*

Response: We agree, and have changed the wording in the introduction, now referring to the "important role" played by risk perceptions in societal responses to climate change (and the lack thereof) (lines 28-29).

*Secondly, if the authors wish to invoke the concept of policy support, then I would encourage them to explain the interrelationships between policy adaption, policy support, and public perceptions of climate change more clearly while citing recent sources such as Yeganeh et al. (2020) and Bergquist et al. (2022)."*

Response: Thank you for pointing us towards these relevant publications. We have added a sentence clarifying our understanding of the relationships between risk perceptions, policy support and policy adoption, referencing Drews and van den Bergh (2015) in addition to Yeganeh et al. (2020) and Bergquist et al. (2022) (lines 29-31).

*"Lines 30-33: "[Lenton et al. (2008] argued that the persistent lack of climate urgency, i.e., insufficiently high-risk perceptions, stems from a "false sense of security" (p. 1792) founded in smooth, gradual projections of climate change." Lenton et al. (2008) merely say that "Society may [!] be lulled into a false sense of security by smooth projections of global change" (p. 1792). The 'may' should be included in the indirect citation as well -> may stem from… Aside from this, I encourage the authors to be more careful when invoking concepts such as risk and urgency (not to equate them). A person who perceives climate change as a significant threat may not necessarily perceive it as an urgent threat. Communicating the risks associated with climate tipping points may affect both types of perceptions. This could be further explored in the introduction or discussion section."*

Response: Good points, and we have edited our reference to Lenton at al. (2008) that "society may be lulled into a false sense of security by smooth projections of global change" as we do not wish to miscommunicate the message (lines 32-33). In our revised version, we have paid careful attention to not equate risk (severity) and urgency, acknowledging that they are related but distinct and may be perceived differently by individuals.

Literature review

*"Sections 2.1 and 2.2 contain a lot of information, not all of which is relevant to the focus of the present study. The sections could be shortened to create a more consistent and clear narrative structure. For instance, when the tipping point risk characteristics are explained, the authors should focus on the main points; each of the characteristics could be explained in only three or four sentences. To structure this section, it could also help to first highlight that the reorganization/state shift is the focal process and the remaining characteristics merely describe how this process unfolds (i.e., in a nonlinear, potentially abrupt fashion) and what it entails (i.e., potential for severe, long-term impacts…)"*

Response: We agree with this assessment (as do other reviewers) and have accordingly condensed the literature review provided in section 2.1. In 2.2, we have shortened the description of risk-relevant characteristics of tipping processes. However, in some cases, we found it was necessary to add a sentence or two to provide previously missing definitions (e.g., of positive feedback in lines 139-143). We have also reorganized this section, starting with a continuous discussion of the four main characteristics of tipping processes, which is followed by a description of additional features that could affect risk perceptions (e.g., the severity of potential impacts).

*"This brings me to the next point: there is not a clear consensus as to whether 'irreversibility' should be considered a CTP characteristic (e.g., Armstrong McKay et al., 2022). After all, irreversibility is a high standard that is extremely difficult to prove. The authors acknowledge this in some parts of the manuscript when they use alternative terms such as 'limited reversibility' and 'irreversible at human timescales'. For reasons of consistency, I would choose a single term, perhaps limited reversibility (or another term that carries the same meaning). From a risk analysis perspective, this can also be framed as a question of controllability: once a tipping point has been crossed, it becomes much more difficult for humans to control exactly how a natural system operates. This is because the regime shift comes with a certain degree of stability/persistence."*

Response: This is a very important observation, in which we fully agree. Also, the issue of reversibility or not is a matter of time, and we highlight this by referring to "timescales that are relevant for humans" on line 157. To address this issue, we are using the term "limited reversibility" throughout the manuscript and specifically refer to human-relevant timescales, and we explain what this means in section 2.2 (lines 152-160).

*"Section 2.1: The authors may want to engage with more recent publications from the field of public perceptions of climate change. Importantly, some of these publications show that in many countries climate change is not perceived as a distant risk anymore (e.g., van Valkengoed et al., 2023; I strongly recommend this review)."*

Response: We are very thankful to the reviewer for highlighting such relevant literature. We have expanded our section on the field of public perceptions of climate change, specifically adding a section reflecting the decline of temporal distancing in public perceptions of climate change, referencing van Valkengoed et al., 2023 (lines 92-94).

*"Sections 2.1-2.3: The summary of the previous literature contains statements that are imprecise and potentially misleading. For instance, the authors state that "[Bellamy and Hulme (2011)] found that climate tipping points increased concern only among participants with an egalitarian value set […]" (lines 196-198). The study that is cited here is a cross-sectional study, not an experimental study. Bellamy and Hulme (2011) merely found that egalitarians were most concerned about climate tipping points. It cannot be concluded that there was an 'increase' in concern or any other variable."*

Response: We greatly appreciate the Reviewer's in-depth knowledge regarding the referenced studies. We have made several corrections and clarifications in this section, e.g., stating that Bellamy and Hulme (2011) "found that concern about climate tipping points was higher among participants with an egalitarian value set while also generating a fatalistic narrative among study participants" (lines 207-208).

*"In their short summary of the study by van Beek et al. (2022), the authors write that "[van Beek et al. (2022)] observed an increase in concern and perceived seriousness of climate tipping points" (line 190f). However, it should be noted that the changes that van Beek et al. (2022) registered on their quantitative measures of concern and seriousness were nonsignificant. That is, the present summary of van Beek et al. (2022) seems to disregard the quantitative findings and only reports the results from the qualitative analysis."*

Response: In our reading of van Beek et al. (2022), they concluded that the qualitative findings outweighed the quantitative ones due to limitations of the quantitative analysis and potential ceiling effects. Hence, our literature review referred to their qualitative findings. However, we have now provided a more detailed report on this paper, which includes a mention of the null-finding in the quantitative component of their analysis (lines 197-202).

*"Line 204f: "[…] a study by Formanski et al. (2022) found no difference between climate risk perceptions related to linear versus non-linear climate change". Given that the present study is in many ways similar to the Formanski et al. (2022) study – e.g., in terms of the design and the dependent variables (for RQ-2) – it would make sense to briefly describe the methodological approach and explain the main findings of this previous study. In general, I encourage the authors to pay particular attention to the most recent publications in this field that directly relate to their research questions – e.g., Bellamy (2023, -> RQ-1) and Formanski et al. (2022, -> RQ-2)."*

Response: We agree with the Reviewer regarding the relevance of the two studies by Bellamy (2023) and Formanski et al. (2022), and the value of a more detailed comparison of their approach and findings with ours. Here, we have expanded our section detailing the study conducted by Formanski et al. (2022), providing a comparison of key methodological similarities and differences with implications for the interpretation of our findings (lines 210-216). We also added a comment on this in the methods section (lines 287-290).

Methodology

*"In part 1a, all participants were asked a series of questions about their climate change risk perceptions" (line 256f). The authors need to provide a list of all questions that were asked here, as well as an explanation for how the responses to these questions were processed/aggregated. The authors also need to disclose all questions that were part of the "tipping point knowledge test".*

Response: All survey questions about risk perceptions can now be found in the supplementary material in Appendix B - questions 1 - 3 pre-intervention and questions 6 - 8 post-intervention. The questions in the climate tipping points knowledge test have now been included in the supplementary material in Appendix B question 4. A description of how the survey responses were processed has now been added (lines 294-295).

*In the results section, the authors present statistics about which tipping point characteristics were most frequently identified. However, from the method section, it is not clear which question stimulated these answers. Section 3.1 only states that participants were asked to name an example of a CTP."*

Response: We have now clarified in section 3.2 (Analysis) that the responses to the prompt to provide an example of a climate tipping point were coded to distinguish groups of participants with different levels of knowledge (lines 322-325). We expand that some participants provided examples of climate tipping elements while others provided characteristics and why we found this meaningful data despite characteristics not being prompted by our question (lines 325-327).

*"Appendix A – which is said to present the stimulus materials (i.e., information packages) – is missing in the document."*

Response: We apologize for this oversight. The stimulus materials have now been included in the supplementary materials to the revised submission in Appendix A.

*"Sample composition: "A nationally representative sample was recruited […]" (line 275f). The authors need to specify in which sense the sample is 'representative'. To me, it looks like the sample is a quota sample, not a probabilistic sample. That would, however, mean that the sample is only representative of the Norwegian general population in terms of selected demographic characteristics (e.g., sex, age, region). If that is the case, the authors need to state this explicitly and provide the quota plan."*

Response: Our survey was performed in close collaboration with a professional polling company which use standard criteria for representative recruitment. We have however expanded the description of the study participants to reflect that our sample was a quota sample to recognise there are members of the Norwegian public that may be overlooked in our study (lines 301-305). We hope that our description now more clearly communicates the selected demographic characteristics.

*"Data analysis (knowledge): Is a qualitative categorization necessary? The authors could just present the familiarity ratings and the frequencies for correct/incorrect CTP examples – and then they could probably draw the same conclusions from this data. Yet, this would not require the presentation of a category system. However, if the authors still wish to use a categorization procedure, then I would advise them to avoid the category label "no knowledge" and to use "no demonstrated knowledge" (see Figure 3, p. 13) instead."*

Response: While a categorization might not be necessary, we believe it is more insightful and better reflects the complex cognitive reality of 'knowing' than merely providing frequencies for correct and incorrect examples. However, we have adjusted the first category label to "no demonstrated knowledge" since we cannot conclusively state that those participants have "no knowledge" and this is now consistent throughout the manuscript. We have also expanded that there are likely several possible explanations for why they did not provide written evidence of their potential familiarity with the concept (lines 315-318).

*"Data analysis (risk perception): Instead of conducting separate independent-sample t-tests on the test scores for t1 and t2, it would be better to conduct an ANCOVA on the post-test scores, with the pre-test scores as a covariate. This is an elegant way to test whether there are differences between the experimental groups at the post-test stage while taking the pre-test scores into account. The paired-sample t-tests (from line 383 onwards) can then be presented as simple 'follow-up analyses'."*

Response: We conducted an ANCOVA analysis on the post-test scores in order to strengthen the validity of our findings in the risk perception analysis and have included the results in this section. We have also included standard deviations for each experimental condition, as well as exact p-values and standardized effect sizes in the manuscript (lines 408-418).

Results

*"Section 4.1, 4.4: The authors should try to meet the journal article reporting standards for empirical research articles in psychology and social sciences. That is, mean values and standard deviations should be reported for each experimental condition (e.g., in a table), as well as exact p-values and standardized effect sizes."*

Response: We have included a table which includes the mean values and standard deviations should be reported for each experimental condition and the exact *p*-values for the ANCOVA and t-tests are now included in the text (lines 408-418).

*"Section 4.1: The authors find that only a few survey respondents rated the information on climate tipping points as "new to them" (see lines 339-341). It is concluded that this indicates socially desirable responding. However, the authors should keep in mind that the free recall of memorized information is generally more difficult than the recognition of memorized information. It is possible that many survey respondents had previously heard of specific tipping elements but were unable to recall that information during the free recall task ("name an example…"). The authors should therefore not simply dismiss their findings on this measure as socially desirable responding. While social desirability could certainly play a role, the present findings could also be an indication that laypeople may be more aware of CTPs and the consequences of unmitigated climate change than researchers often assume (even though that CTP knowledge might not be highly accessible, as it was only activated through the confrontation with specific stimulus materials). The authors should also consider the possibility that many laypeople may be aware of the catastrophic consequences of unmitigated climate change, but they may not associate these impacts with the concept of tipping points or simply do not know the relevant terminology. This tends to be a principal weakness of studies that only ask people 'whether they have heard of climate tipping points'. Thus, the present results give us limited insights into laypeople's expectations about how climate change will unfold in the future. This is yet another reason why the authors need to be cautious in their interpretation of the present results."*

Response: The Reviewer makes an important argument, which we now discuss in section 4.1 (lines 373-379). Our methodological approach differs from Bellamy's design regarding the role of free recall, and this deserved some attention.
The Reviewer's comments regarding laypeople's underexplored general awareness of the potentially catastrophic impacts of unmitigated climate change are apt. However, we did not seek to investigate these general beliefs about the nature of future climate impacts; we were interested specifically in the effect of knowledge about tipping points on climate risk perceptions. While we do not discuss this issue in our manuscript, future research should explore this phenomenon.

Discussion

*"Lines 418-421: "Our results contrast with recent findings by Formanski et al. (2022) […]. One explanation for this difference might be that Formanski et al focused on a single characteristic of tipping points (non-linearity), which might not be the feature that generates most concern". The treatment materials by Formanski et al. (2022) also highlighted another feature – severe impacts. However, it is true that other features such as the limited reversibility of the impacts were not explicitly mentioned, which is a valuable observation. At*

*the same time, the authors could also consider the fact that there were other differences in terms of the specificity, scope, and length of the materials - Formanski et al. (2022) acknowledged that the length and simplicity of the materials could explain their null findings. A simple reframing of climate change as a dynamic phenomenon (with the help of the CTP concept) may not be enough to increase concern; yet the current study indicates that a more elaborate discussion of CTPs and the associated risk characteristics could help!"*

Response: We have expanded our discussion of why our results differ from Formanski et al. (2022), why this might be, and how future research could address this more thoroughly. As mentioned above (lines 477-485), we also provide more detail on how this study compares to ours in the methods section.

*"Lines 426-429: "We did not observe any effect of information of climate tipping points on beliefs about whether or not it is too late to act on climate change. This could be attributed to the public's tendency to downplay the seriousness of these risks due to certain cognitive biases, and that systematic risk associated with climate tipping points pose unique learning challenges that is not easily grasped by participants". First of all, the authors could point out that the null finding on this item is consistent with the results that Formanski et al. (2022) report for the dependent variable 'efficacy beliefs'. Secondly, the explanation that the authors offer for the null finding would only seem plausible to me if the text presented to participants had, in some way, suggested that the crossing of multiple CTPs is inevitable – why else would they be prompted to believe that it is too late to act on climate change?"*

Response: The Reviewer's point made us reconsider the inclusion of this finding and argument. We decided to remove this section, and no longer present our results on our participants' response to "whether or not it is too late to act on climate change".

Conclusion

*"The excurse on social tipping points (from line 475 onwards) comes a bit out of nowhere. The authors may want to consider deleting this part or embed it more deliberately, so that it fits into the conclusion."*

Response: We have removed the discussion of social tipping points from our conclusion. While we find them relevant to the conversation on climate tipping points and public risk perceptions of climate change, this was not discussed or elaborated on in the body of our work, therefore does not have an obvious place in the conclusion.

Technical corrections

*"Stoddard (2021)" is missing in the references"*

Response: we have added Stoddard (2021) and checked (and corrected) other errors with regards to the references.

*"Some publications that are listed in the references section are not cited in the text (e.g., Trope & Liberman, 2010)"*

Response: We have screened the manuscript for errors relating the references.

*"Authors names -> correct spelling errors (e.g., Russil -> Russill)"*

Response: (Line 188) We have corrected this error and screeded the manuscript for other such errors.

*"Throughout the manuscript the authors frequently use phrases such as "could be affected by climate tipping points" (line 58). And when they introduce the concept in line 35, they state that "Climate tipping points refer to dynamics in the Earth system […]". Technically, tipping points are not 'dynamics' within the climate system – they are thresholds; hence, tipping points do not 'affect' countries – only the impacts that the crossing of tipping points has can affect countries. I presume that the authors mean the processes that are initiated by tipping events, not the 'tipping point' itself?"*

Response: We have been clearer in our terminology with regards to the difference between a tipping point and the change process associated with the passing of a tipping point throughout the manuscript. We have also corrected the statement that climate tipping points are not dynamics, but thresholds in the Earth system in order to be as clear as possible when distinguishing climate tipping points from the impacts and processes they set in motion.

*"The authors use formulations such as "the effects of climate tipping points on" (line 70). I presume that the authors mean the effects of the 'presentation of' or 'exposure to' information on climate tipping points?"*

Response: We do indeed refer to the effects of the exposure to knowledge or information about climate tipping points; we have corrected this instance (line 70) and other such instances in the manuscript.

*"Line 323: "50% indicated little or no familiarity" – in case two categories were combined here, the authors should provide frequencies for each category ("little familiarity" and "no familiarity")"*

Response: The frequencies have now been provided for each category (lines 352-354).

**Reviewer 2**

*"The paper provides excellent background on risk perceptions of climate change in general and climate tipping points, as well as on the Norwegian context for the study. The section on risk-relevant characteristics of tipping points, however, seems a little too long. The main points could be made more succinctly."*

Response: We acknowledge this very positive judgement of our manuscript. The section on risk-relevant characteristics was also highlighted by Reviewer 1, and we have accordingly now significantly reduced the amount of information in section 2.2 in order to focus more clearly on the key messages.

*"At the same time, this section also seems a little too prescriptive. It is rightly noted that definitions of climate tipping points are varied, but it would be worth unpacking the divergences in more detail and adopting either a more flexible understanding of the concept,*

*or alternatively being prescriptive about only those features that are common to all climate tipping points. For example, irreversibility is identified as one of the key features (and by the public themselves), but this is in fact not a feature of many climate tipping points, e.g., Arctic sea ice, Atlantic Meridional Overturning Circulation, ocean acidification, etc. So, it can be contested as to whether identifying 'irreversibility' would contribute towards a 'correct' understanding of climate tipping points. In other words, it would be good to see some reflection on the problematic nature of 'knowledge' about climate tipping points, and what the implications are for the present (and future) research."*

Response: This is a core point, also commented on by Reviewer 1, and the issue of irreversibility (now referred to limited reversibility) has been elaborated more (lines 152-157). This qualification no doubt also depends on the post-tipping time horizon. It was not our intent to come across as prescriptive in this section, but to identify the risk-relevant features common to climate tipping points in the literature (and based on emerging agreement on tipping point definitions in recent reviews such as Armstrong McKay et al., 2022 and Milkoreit et al., 2018 which both identify limited reversibility as a feature of climate tipping points). We have been more specific that we have chosen to focus on only certain characteristics, recognising the divergencies while stating our focus (lines 126-129).

*"A more minor point is that in some places, notably lines 100 to 106, there are repetitions of the text."*

Response: We have corrected this error in the text and removed the repetitions.

**Reviewer 3**

*"With regards to the introduction, it is quite hard to discern what is the research question that the study is exploring. It only comes until line 45-55 (4th and 5th paragraph) where readers finally understand the intention of the manuscript. The introduction would significantly benefit if the research question were brought earlier on in the first paragraph and then provide information on what has been done and why it is important to explore climate risk perceptions of tipping points from the public in Norway."*

Response: We agree that it is important to arrive at the research questions early on and have revised the text accordingly by the research question earlier into the introduction (lines 36-37).

*"There are three issues I would like to raise on the literature review on climate risk perceptions (section 2). First, I would like to suggest including a brief overview of the literature that studies affective dimensions of climate risk perceptions, as this is another growing field in the literature that significantly influences how individuals perceive climate risks."*

Response: We have thoroughly considered this, but decided not include a section addressing the literature that studies affective dimensions of climate risk perceptions, as our study does not explore this, and the text is already quite extensive. As the Reviewer notes, the literature review should focus on the specific topic of the research, and we did not include affective or emotional variables in our work. However, we understand that it is an

important area of research and should be considered in future research when investigating risk perceptions of climate change.

*"Second, I don't understand the difference between section 2.1 and 2.2 that both talk about risk perceptions (it may be a typo). This needs attention.*

Response: Apologies and yes, this error in the headlines has been addressed in the manuscript and corrected. Section 2.1 is about general climate risk perception scholarship; section 2.1 concerns the risk-relevant characteristics of climate tipping points.

*"Three, section 2.2 "Risk-relevant Characteristics of Climate Tipping Points" and 2.3 "Perceptions of Climate Tipping Point Risk" are far too long and can be summarised. I think it's important to remember that the study is about the public in Norway and how they perceive climate risks about tipping points. The literature review and background information should provide enough context for this, so this section can significantly be summarised to be more specific about what readers need to know (think about the audience for this piece and what they may already know)."*

Response: This is an important observation made my all the reviewers. We have shortened sections 2.2 and 2.3 in order to communicate the information more concisely, which is followed section 2.3 which discusses the significance of Norway as a case study.

*"With regards to the description of the study participants, it would be interesting to provide more information (if possible) to know which audience may have not been able to be represented with this method. It is important to be clear about the limitations of our studies, and this is one where it is likely that some publics within Norway remained invisible (e.g., immigrants, international students, refugees, people with some disability, non-binary, etc.). Also, what is the level of study from study participants?"*

Response: This is a very important comment which should be considered in all studies that seek to provide a nationally representative sample of a population. We have been more transparent on the limitations of our study where it comes to accurate representation of the Norwegian population (lines 301-305). It is important to recognise members within a population that remain invisible using our methodology and a limitation within our study (lines 304-305).

*"Where the results of the t-tests independently verified? And if so, how?"*

Response: We used a *t*-test for independent means which has been clarified in the text. We cannot state that our results were verified by an unbiased and impartial third party, however, we conducted two statistical tests and yielded statistically significant results in both tests (ANCOVA as recommended by Reviewer 1 and *t*-test for independent means). However, this is a limitation in our study and is now declared in our manuscript as such (lines 480-481).

*"Finally, on the conclusions, the manuscript fails to provide meaningful recommendations to other potential audience of this study (e.g., government and academia). If the authors believe that the public in Norway need to know about tipping points, then it would be important to explain this. In other words, the conclusion should describe why this knowledge*

*is necessary and what is needed or recommended to disseminate this knowledge. For example, if people in Norway understand the risks of tipping points, could we expect some sort of change? In what way?*

Response: We agree and have included recommendations in sections 5 and 6 in the latest manuscript submission which we believe elaborates on recommendations and expanded on why knowledge on climate tipping points may be important to the public.

*"[…] This takes me to my next comment. The manuscript should provide future research opportunities and also be clear about the limitations of the study."*

Response: We have expanded upon future research opportunities in the discussion and conclusion (sections 5 and 6), and also been clearer about the limitations of our study.

Minor comments

*"Page 1 line 25: Incorrect factual statements are made in the introduction when referring to "internationally agreed-upon targets". Nationally determined contributions are the opposite of internationally agreed. The Paris Agreement is based on a bottom-up self-determined approach."*

Response: Thank you for highlighting this error in our wording regarding the Paris Agreement. We have changed the phrasing (Line 26) to reflect the fact that the Paris Agreement contains global goals that have been internationally agreed, but no (emission) targets for individual countries.

All of these changes to the manuscript can be found in the marked-up copy which has been included with our resubmission.

I sincerely hope that these modifications align with the expectations of the reviewers and the editor. I am confident that the revisions have strengthened the overall quality of the manuscript.

On behalf of myself and my fellow authors, I thank you again for considering our manuscript and look forward to your feedback and decision regarding our research paper. Please do not hesitate to contact me if further clarification or additional information is required.

Kind regards,

Christina Nadeau

---

## Referee Report (RR1)

Referee Report: Nadeau et al. (2024) – esd-2023-23
referee report
en

**Referee Report: Nadeau et al. (2024) – esd-2023-23**

**1. General comments**

In the revision process, the authors have made several important changes that have improved the quality of the manuscript. The introduction is now more concise, the methods section is more detailed and informative. The materials that are now provided make it easier to understand and reconstruct the methodology. However, now that the methodology is properly disclosed, some notable flaws have also become evident. Specifically, I have identified one major issue and multiple minor issues that primarily relate to the processing and interpretation of the collected survey data.

**2. Specific comments**

1) Section 4.1 & 4.2: Strictly speaking, the authors only used two questions to assess the "general state of knowledge of climate tipping points" in the Norwegian population: 1. "How familiar are you with the CTP concept?", and 2. "Can you give an example of a CTP? If yes: _________ ". This is not a comprehensive 'CTP knowledge test'. The authors did not specifically ask for a definition of the concept or a description of CTP characteristics. Nevertheless, the authors try to draw conclusions about laypeople's awareness of CTP characteristics from their answers to the second question, because it turned out that some participants (n = 161) submitted general comments on the CTP concept/ CTP characteristics. It is not entirely clear why that happened. It seems that many participants were unable to provide specific CTP examples, or they simply misunderstood the question. In any case, these general comments should not be used to draw statistical conclusions about laypeople's awareness of CTP characteristics. The comments are only the side product of an open-ended question that focused on a different issue. The authors should note that the frequencies displayed in Table 1 do not include the potential responses of the roughly 700 participants who chose not to provide general comments. Some of these participants might have been able to write something about CTP characteristics if they had been asked about this specifically. And these individuals might have had other characteristics in mind than the remaining participants. Thus, I would strongly advise the authors to disregard the general comments about CTP characteristics and focus only on the question of whether or not participants were able to recall a CTP example in response to this prompt question.

From a methodological perspective, this should then also preclude any broad conclusions about CTP knowledge in the general population of Norway. After all, the authors only asked for familiarity ratings and examples for CTPs. A categorization of participants into different 'CTP knowledge categories' seems inappropriate. Instead, the most important results of the survey can be summarized in three short paragraphs: A) one paragraph describing the familiarity ratings, B) one paragraph stating how many participants were able to provide a correct CTP example vs. how many were unable to do so/provided general comments, and C) a paragraph outlining which CTPs were mentioned most frequently. With this in mind, I would strongly encourage the authors to remove section 4.2, rewrite section 4.1, and reformulate their conclusions regarding the state of public knowledge about CTPs in Norway (see discussion).

**Minor issues**

2) Section 4.4: The authors still do not provide standardized effect sizes even though these are essential for the interpretation of the treatment effect. From the mean values and standard deviations, it can be derived that the effect that was observed here is really small (Cohen's $d = 0.08$ for the pre-post difference in Group A, with SD(y) in the denominator of d; note that in the social sciences, $d = 0.2$ constitutes a small effect, $d = 0.5$ constitutes a medium effect, and $d = 0.8$ constitutes a large effect; see Cohen, 1988). This should be explicitly acknowledged and discussed in the manuscript.

In this context, I would like to draw the authors' attention to the fact that the study by Formanski et al. (2022) found a difference of $d = 0.04$ between their experimental condition (non-linear climate change portrayal) and their reference condition (linear climate change portrayal). But that difference was nonsignificant, perhaps because the sample size used in that study ($N = 360$) was too small to detect such a tiny effect. Nonetheless, it can be seen that the effect size obtained in that previous study was not so different from the effect size observed in the present study, which used a large sample of $N = 851$. In large samples, even small effects become statistically significant. Given these considerations, I would not flatly conclude that the findings of the present study "contrast" with the results of Formanski et al. (see line 477) – that study concluded that such an effect might not exist – but if it exists, it is likely to be small. The present study now provides evidence that this effect could indeed exist, but that it is likely to be small (= the conclusions are not entirely contradictory).

3) Section 4.4: The present study also found that exposure to CTP information did not influence fatality ratings ("Is it too late to do anything about climate change?"), which is essentially congruent with what Formanski et al. (2022) found on their efficacy beliefs measure. This result is not mentioned anymore in the revised version of the manuscript, even though a) the item is mentioned in the methods section and b) the introduction raises the (very important) question of whether exposure to information on CTPs induces fatalism/reduces efficacy beliefs. I believe that this null finding is informative and that it should be described and discussed in the manuscript. I would just like to note that I am skeptical of the explanation the authors initially proposed for this finding. In the first draft, the finding was attributed to "the public's tendency to downplay the seriousness of these risks due to certain cognitive biases […]" – but that would have only made sense if the materials had, in some way, suggested that crossing climate tipping points is inevitable. A more plausible and straightforward explanation for this null finding could be that exposure to information about CTPs may not necessarily promote fatalism.

4) Section 3.1: The authors should clarify which items were used to measure climate change risk perceptions (CCRP). The keywords they list in line 273 do not match the questions q1r1-q1r3 in the appendix. In addition, the authors should follow common reporting conventions and provide a reliability estimate for their CCRP scale (e.g., McDonald's Omega).

5) Section 4.4: In the context of randomized controlled trials with a pre-post measurement, the ANCOVA technique is only used to compare the mean post-test scores across the experimental conditions (Group A vs. Group B), while including the pre-test scores (here: CCRP at t0) as a covariate in the model (e.g., Frison & Pocock, 1992). What this means here is that the ANCOVA should replace the independent sample t-test for the post-test scores (lines 419-421), because the ANCOVA is simply more informative as it also takes the pre-test scores into account. The paired-sample t-tests (lines 430-433) can still be presented as a follow-up analyses – these tests provide information that is not directly uncovered by the ANCOVA. The analyses that are currently presented in the first paragraph of section 4.4 are either irrelevant to the research question (--> effect of time averaged across the two conditions, see lines 409f), or already reported in the text (difference between the groups at t0, see line 412 vs. lines 427f), or not precisely described (line 411 – is this the result of the ANCOVA that is announced in the second sentence of the paragraph?).

6) Lines 105f: "Some studies have demonstrated that instruction, information, and knowledge about climate change increase climate risk perceptions (Aksit et al., 2018; Milfont, 2012; van der

Linden, 2015; Xie et al., 2019)" – Most of the studies cited here are only correlational studies, which is why the authors might want to rephrase this sentence; see also line 244 – same issue here

7) Lines 301-305: The main drawback of the sample is that it is not a probabilistic (random) sample. This means that the sample composition could differ from the composition of the general population in terms of relevant characteristics that were not considered in the quota plan (e.g., education, income, social status, personality traits…).

8) Lines 480f: "Our results were not independently verified by an unbiased and impartial third party, which is a limitation of our study." – Do the authors mean that the statistical analyses were not re-run by a third party? As far as I know, this is not common practice in social sciences studies. What is more common is that the data set obtained in a given study is made publicly accessible (in an anonymized form), so that everyone can re-run the analyses.

**3. Technical corrections**

1) Milfont (2012) is missing in the reference list

2) Line 167: "the likelihood of triggering climate tipping points is "dangerously close"…" This sentence should be rephrased.

3) Lines 170f: "e.g. shift in turbid and clear-water phase in lakes" – I would advise the authors to only cite examples of climate tipping elements here, to avoid confusion.

4) Lines 119-125: This paragraph could be removed to reduce the length of the article.

5) Lines 63-68: This part could be shortened – some of this information is repeated in section 2.4.

6) Line 407: "Effect of climate tipping points on Level of Concern for Climate Change", should be changed to "effect of information about…/ effect of exposure to information about…"

7) The results of F-tests and t-test should be reported in a consistent format throughout the manuscript (e.g., APA format).

8) In its current form, Appendix B lacks structure and contains several unclear phrases, e.g., "Your local environment - When do you think the climate crisis will start to affect the following?" – here, 'your local environment' should be placed after the question.

---

## Author Response (AR2)

**Author Response: Nadeau et al. (2024) – esd-2023-23**

1.  **General comments**

In the revision process, the authors have made several important changes that have improved the quality of the manuscript. The introduction is now more concise, the methods section is more detailed and informative. The materials that are now provided make it easier to understand and reconstruct the methodology. However, now that the methodology is properly disclosed, some notable flaws have also become evident. Specifically, I have identified one major issue and multiple minor issues that primarily relate to the processing and interpretation of the collected survey data.

**Response:** We would like to thank the reviewer for taking the time to provide thorough and valuable feedback on our revised manuscript. We appreciate the positive comments on the improvements. We acknowledge the thorough response with constructive suggestions. While we have adhered to most of these, there are cases where we also disagree, and below we have addressed the major and minor comments in detail.

2.  **Specific comments**

1) Section 4.1 & 4.2: Strictly speaking, the authors only used two questions to assess the "general state of knowledge of climate tipping points" in the Norwegian population: 1. "How familiar are you with the CTP concept?", and 2. "Can you give an example of a CTP? If yes: _________ ". This is not a comprehensive 'CTP knowledge test'. The authors did not specifically ask for a definition of the concept or a description of CTP characteristics. Nevertheless, the authors try to draw conclusions about laypeople's awareness of CTP characteristics from their answers to the second question, because it turned out that some participants (n = 161) submitted general comments on the CTP concept/ CTP characteristics. It is not entirely clear why that happened. It seems that many participants were unable to provide specific CTP examples, or they simply misunderstood the question. In any case, these general comments should not be used to draw statistical conclusions about laypeople's awareness of CTP characteristics. The comments are only the side product of an open-ended question that focused on a different issue. The authors should note that the frequencies displayed in Table 1 do not include the potential responses of the roughly 700 participants who chose not to provide general comments. Some of these participants might have been able to write something about CTP characteristics if they had been asked about this specifically. And these individuals might have had other characteristics in mind than the remaining participants. Thus, I would strongly advise the authors to disregard the general comments about CTP characteristics and focus only on the question of whether or not participants were able to recall a CTP example in response to this prompt question.

**Response:** The Reviewer is concerned with our interpretation of the survey results regarding knowledge, i.e., what kind of interpretations our data allows, and what constitutes a comprehensive knowledge test. We do not claim to have provided a comprehensive

knowledge test, but an assessment of awareness and understanding that both recognizes the complex nature of the variable we label 'knowledge' and provides meaningful insights about this variable for a national population, i.e., captures what is called 'public understanding of science'.

Knowledge can be defined as cognitive processes that relate to awareness or familiarity with information that is believed to be factual. Different types of knowledge (e.g., scientific, elite or decision-maker, and public) and different degrees of understanding can be distinguished, e.g., abstract-conceptual knowledge to usable-practical knowledge. And there are various challenges related to the display/demonstration of knowledge, e.g., recall/memory vs. active generation of information. In our case, we are studying public knowledge of a scientific concept. In this context, the expectation is to gauge general familiarity with a phenomenon without necessarily expecting a study participant to provide a scientific definition of the concept.

In the case of climate tipping points, this methodological challenge - eliciting responses that demonstrate knowledge - is compounded by the fact that multiple scientific definitions exist, with some ongoing disagreement about the necessary conditions for a tipping process. In many ways, definitions depend on context and scientific discipline. Again, it would not be reasonable to expect members of the public to provide a definition, which is why our survey questions did not seek to elicit this form of knowledge demonstration. Instead, we opted for a familiarity rating on a scale, which is a common approach (Ladwig et al. 2012).

> Ladwig, P., Dalrymple, K. E., Brossard, D., Scheufele, D. A., & Corley, E. A. (2012). Perceived familiarity or factual knowledge? Comparing operationalizations of scientific understanding. Science and Public Policy, 39(6), 761-774.

Further, examples are key for learning about a novel concept. Major cognitive theories (prototype and exemplar theories) (e.g., Park 2013) explain knowledge acquisition (i.e., learning a new concept) in terms of category development, which relies on examples. While the definition of tipping points remains unstable, there is broad agreement among scientists regarding examples of climate tipping points. All media reporting on the topic - the key source of public knowledge - also tends to focus on examples. For example:

- https://www.nytimes.com/2024/02/15/climate/tipping-points-for-the-planet.html
- https://www.nytimes.com/2023/07/25/climate/atlantic-ocean-tipping-point.html
- https://grist.org/climate-tipping-points-amazon-greenland-boreal-forest/
- https://www.carbonbrief.org/explainer-nine-tipping-points-that-could-be-triggered-by-climate-change/
- https://www.klimastiftelsen.no/publikasjoner/vippepunkter-i-klimasystemet
- https://www.forskning.no/klima-klimatiltak-miljopolitikk/forskere-advarer-mot-vippepunkter-i-en-ny-rapport/2300644

Hence, we developed a survey question that combined self-assessment of one's ability to provide an example and actual ability to provide an example.

Park, J. J. (2013). Prototypes, exemplars, and theoretical & applied ethics. *Neuroethics*, *6*, 237-247.

Given this cognitive structure of knowledge, and the specific context of climate tipping points, we deliberately designed the survey to ask in multiple stages of increasing cognitive difficulty about participants' familiarity with climate tipping points, providing the opportunity to reveal how well they (believe they) know the phenomenon. The question sequence allows participants to first indicate their degree of self-assessed knowledge (familiarity) with a Likert scale without the need for active recall. For those participants who indicated some familiarity with the concept, i.e., could be expected to have some ability for active recall, this was followed by a question that gauged the self-assessed ability to recall an example. This was considered an easier cognitive task than providing a definition. If a person indicated that they were not able to provide an example, it is of course possible, but very unlikely, that they would have been able to offer a definition or characteristics of tipping point if specifically asked for those.

Participants who answered yes to this question (ability to provide an example) were invited to provide an example, and chose to respond in three different ways: (i) not to provide an example or any other information, (ii) to provide (what they thought was) an example, or (iii) to provide descriptions of the phenomenon instead of an example. While the reviewer suggests it is 'unclear why this happened', we do not think this is the case. Given the survey structure, these response types are expected. We argue that these responses can be interpreted regarding the respondents' state of knowledge:

- No response – The person experienced difficulty in recall and recognized that their self-assessment in the prior question had been too high. They might have some knowledge, but were not able to demonstrate it.
- Example – Demonstration of some knowledge, which can include instances of events that are not climate tipping points, i.e., misconceptions that are cases of incorrect knowledge;
- Broader, descriptive comments – The person has some knowledge but is not able to give an example (i.e., over assessed knowledge), and decided to provide a descriptive account or other conceptual associations instead of an example.

The descriptive comments could be coded based on common characteristics of tipping points. While we do not see this as redundant information or even flawed in the sense that it should be disregarded, we agree with the Reviewer that the data cannot be used for statistical analyses regarding all survey participants (n=851) with inferences for the population of Norway since we did not ask all participants to describe characteristics of tipping points. However, it is not unreasonable to use this information to describe the pattern of knowledge displayed by those respondents who saw and answered the open-ended question (n=153), i.e., the group with the most self-assessed knowledge. Hence, we clarified the circumstances and limitations of data collection and emphasised that our descriptive statistics do not necessarily apply to the whole population. Nevertheless, we have moved Table 1 into the Supplementary

material. We believe that providing this additional information with the necessary clarification about the data elicitation process is interesting and can inform future work, e.g., on the most and least understood features of tipping processes.

We also would like to emphasise that survey design, recruitment and statistical analysis were done in cooperation with a leading polling company in Norway. While we do agree that this does not provide "general state of knowledge of climate tipping points", and hence have revised our manuscript's terminology, we argue that it provides knowledge about the awareness of the concept, and also that the examples serve as a "control" to see if the knowledge self-assessment is justified by adequate examples.

From a methodological perspective, this should then also preclude any broad conclusions about CTP knowledge in the general population of Norway. After all, the authors only asked for familiarity ratings and examples for CTPs. A categorization of participants into different 'CTP knowledge categories' seems inappropriate. Instead, the most important results of the survey can be summarized in three short paragraphs: A) one paragraph describing the familiarity ratings, B) one paragraph stating how many participants were able to provide a correct CTP example vs. how many were unable to do so/provided general comments, and C) a paragraph outlining which CTPs were mentioned most frequently. With this in mind, I would strongly encourage the authors to remove section 4.2, rewrite section 4.1, and reformulate their conclusions regarding the state of public knowledge about CTPs in Norway (see discussion).

**Response:** We disagree with the Reviewer's argument that our data collection method does not allow any broad conclusions about CTP knowledge among the Norwegian public. As outlined above, we devised a scientifically grounded approach to knowledge elicitation among the Norwegian public. We have clarified our assumptions about public knowledge and justified the survey design that avoids asking for a definition or characteristics, but relies on self-assessment (perceptions) of ability with a 'check' for actual ability to recall an example. Other studies, including Bellamy 2023, also assess knowledge ('awareness') without asking participants to provide a definition.

We also argue that separating respondents into these four categories should be appropriate. These kinds of categorizations have some inherent subjectivity, but are a standard approach to characterise the population with distinct categories. We have however adjusted the way we categorise to maximise categorization in favour of knowledge and to remove a differentiation based on familiarity with characteristics (previous distinction between some and good knowledge). This revised categorization likely overestimates the state of knowledge among the public, but accounts for some of the Reviewer's concerns about the nature and interpretation of our data.

1) No knowledge - based on familiarity rating only (know very little, have never heard of climate tipping points) [52%]

2a) Some knowledge but no demonstrated ability to provide an example or unelicited description - based on familiarity rating (categories 1-3), answered NO to question about ability to provide an example OR answered YES to question about ability to provide an example, but did not respond to prompt to provide an example - [25% n=209]

2b) Some knowledge but only demonstrates incorrect knowledge - based on familiarity rating (categories 1-3), answered yes to question about ability to provide an example, but provided a wrong example or wrong unelicited comments [5% n=44]

3) Good knowledge - based on familiarity rating (categories 1-3), answered yes to question about ability to provide an example, provided at least one correct example or unelicited comments with correct characteristics of climate tipping points [13% n=109]

We have modified the text, retained section 4.2 with an expanded explanation of the nature of the data, and placed Table 1 in Supplementary Materials.

**Minor issues**

2) Section 4.4: The authors still do not provide standardized effect sizes even though these are essential for the interpretation of the treatment effect. From the mean values and standard deviations, it can be derived that the effect that was observed here is really small (Cohen's d = 0.08 for the pre-post difference in Group A, with SD(y) in the denominator of d; note that in the social sciences, d = 0.2 constitutes a small effect, d = 0.5 constitutes a medium effect, and d = 0.8 constitutes a large effect; see Cohen, 1988). This should be explicitly acknowledged and discussed in the manuscript.

**Response:** We thank the reviewer for pointing out the missing standardised effect sizes in our text. We ran the analysis again in R and found a standardised effect size observed to be 0.08 standard deviation units - which indeed shows that the effect observed in our data can be interpreted as very small. We have now included this in the manuscript (lines 499-502) and it is discussed (lines 569 - 571). We have also reflected this in the abstract (line 22) and conclusion (line 605).

In this context, I would like to draw the authors' attention to the fact that the study by Formanski et al. (2022) found a difference of d = 0.04 between their experimental condition (non-linear climate change portrayal) and their reference condition (linear climate change portrayal). But that difference was nonsignificant, perhaps because the sample size used in that study (N = 360) was too small to detect such a tiny effect. Nonetheless, it can be seen that the effect size obtained in that previous study was not so different from the effect size observed in the present study, which used a large sample of N = 851. In large samples, even small effects become statistically significant. Given these considerations, I would not flatly conclude that the findings of the present study "contrast" with the results of Formanski et al. (see line 477) – that study concluded that such an effect might not exist – but if it exists, it is

likely to be small. The present study now provides evidence that this effect could indeed exist, but that it is likely to be small (= the conclusions are not entirely contradictory).

**Response:** This is true, it is well known e.g. from epidemiological studies with large samples that significance should be treated with caution, and statistical significance not by necessity implies relevant differences. We have now discussed this in the text, in order not to inflate our findings or the significance of our results (lines 569-571).

3) Section 4.4: The present study also found that exposure to CTP information did not influence fatality ratings ("Is it too late to do anything about climate change?"), which is essentially congruent with what Formanski et al. (2022) found on their efficacy beliefs measure. This result is not mentioned anymore in the revised version of the manuscript, even though a) the item is mentioned in the methods section and b) the introduction raises the (very important) question of whether exposure to information on CTPs induces fatalism/reduces efficacy beliefs. I believe that this null finding is informative and that it should be described and discussed in the manuscript. I would just like to note that I am skeptical of the explanation the authors initially proposed for this finding. In the first draft, the finding was attributed to "the public's tendency to downplay the seriousness of these risks due to certain cognitive biases […]" – but that would have only made sense if the materials had, in some way, suggested that crossing climate tipping points is inevitable. A more plausible and straightforward explanation for this null finding could be that exposure to information about CTPs may not necessarily promote fatalism.

**Response:** We no longer include the fatalism results in this manuscript because it is not central to our research questions, but thank the reviewer for pointing out the instances mentioned in the test. We have removed the text that refers to fatalism.

4) Section 3.1: The authors should clarify which items were used to measure climate change risk perceptions (CCRP). The keywords they list in line 273 do not match the questions q1r1-q1r3 in the appendix. In addition, the authors should follow common reporting conventions and provide a reliability estimate for their CCRP scale (e.g., McDonald's Omega).

**Response:** We have clarified which questions were used to measure climate change risk perceptions in section 3.1. We have also provided a reliability estimate for our CCRP scale: "the calculated value for Cronbach's alpha was $\alpha = 0.897$ indicating strong internal consistency reliability among the questions measuring CCRP, suggesting that the questions are highly correlated and likely measure the same underlying construct effectively". (lines 279-285)

5) Section 4.4: In the context of randomized controlled trials with a pre-post measurement, the ANCOVA technique is only used to compare the mean post-test scores across the experimental conditions (Group A vs. Group B), while including the pre-test scores (here: CCRP at t0) as a covariate in the model (e.g., Frison & Pocock, 1992). What this means here is that the ANCOVA should replace the independent sample t-test for the post-test scores

(lines 419-421), because the ANCOVA is simply more informative as it also takes the pre-test scores into account. The paired-sample t-tests (lines 430-433) can still be presented as a follow-up analyses – these tests provide information that is not directly uncovered by the ANCOVA. The analyses that are currently presented in the first paragraph of section 4.4 are either irrelevant to the research question (--> effect of time averaged across the two conditions, see lines 409f), or already reported in the text (difference between the groups at t0, see line 412 vs. lines 427f), or not precisely described (line 411 – is this the result of the ANCOVA that is announced in the second sentence of the paragraph?).

**Response:** We have rerun the statistical analyses and have rewritten our text to include the ANOVAs and ANCOVA to better communicate our step-by-step tests run in R. All the F, t and p values are shown in the text. We include one of the paired-sample t-tests clearly as follow-up analyses but have removed the specific t-test the reviewer points to as a repetition of the ANCOVA analysis (lines 482 - 517).

6) Lines 105f: "Some studies have demonstrated that instruction, information, and knowledge about climate change increase climate risk perceptions (Aksit et al., 2018; Milfont, 2012; van der Linden, 2015; Xie et al., 2019)" – Most of the studies cited here are only correlational studies, which is why the authors might want to rephrase this sentence; see also line 244 – same issue here

**Response:** We have reworded the sentence to reflect the limitations of statistical analyses more carefully. The sentence now reads "Some studies have shown clear correlations between instruction, information and knowledge on the one hand and climate risk perception on the other" (lines 107-109).

7) Lines 301-305: The main drawback of the sample is that it is not a probabilistic (random) sample. This means that the sample composition could differ from the composition of the general population in terms of relevant characteristics that were not considered in the quota plan (e.g., education, income, social status, personality traits…).

**Response:** We acknowledge the importance of utilising probabilistic (random) sampling methods for the generalisability of our findings to the broader population. As highlighted in your comment, non-probabilistic sampling has implications for the representativeness of the sample, particularly concerning characteristics excluded from the quota plan. However, quota sampling is a common and widely accepted approach in large-scale surveys - quota samples represent the population of interest in a real sense. The traits included in the sampling are key for a successful study. Our quota plan included major characteristics that reflect the demographic makeup of the Norwegian population. As stated above, to ensure the quality of our approach, we worked with one of Norway's best-known polling services with decades of experience. Nevertheless, we have duly noted the limitations associated with our sampling approach in the manuscript, especially when it comes to minorities in the population. We have added an extra sentence to emphasise this limitation (lines 328-330).

8) Lines 480f: "Our results were not independently verified by an unbiased and impartial third party, which is a limitation of our study." – Do the authors mean that the statistical analyses were not re-run by a third party? As far as I know, this is not common practice in social sciences studies. What is more common is that the data set obtained in a given study is made publicly accessible (in an anonymized form), so that everyone can re-run the analyses.

**Response:** When rerunning our analysis, we also had our results verified by a third party.

**3. Technical corrections**

1) Milfont (2012) is missing in the reference list
**Response:** Milfornt (2012) is now included in the reference list.

2) Line 167: "the likelihood of triggering climate tipping points is "dangerously close"…"
This sentence should be rephrased.
**Response:** We have rephrased this sentence to "the risk of triggering some climate tipping points may be "dangerously close".

3) Lines 170f: "e.g. shift in turbid and clear-water phase in lakes" – I would advise the authors to only cite examples of climate tipping elements here, to avoid confusion.
**Response:** The intention of this example was to highlight the range of timescales relevant to tipping elements, however, we recognise that our paper focuses on climate tipping elements and therefore using this particular example may cause confusion. We have therefore removed this example from the manuscript and instead focused on timescales relevant to climate tipping elements.

4) Lines 119-125: This paragraph could be removed to reduce the length of the article.
**Response:** We have removed part of this paragraph; however, we have retained the first two sentences in order to highlight how the scholarship on climate tipping points differs from general climate change scholarship historically.

5) Lines 63-68: This part could be shortened – some of this information is repeated in section 2.4.
**Response:** This section has been shortened in order to limit repetition in the article (lines 68-71).

6) Line 407: "Effect of climate tipping points on Level of Concern for Climate Change", should be changed to "effect of information about…/ effect of exposure to information about…"
**Response:** Thanks to the reviewer for pointing this out, we have changed this to "Effect of Information about Climate Tipping Points on Level of Concern for Climate Change" to be in line with our article title.

7) The results of F-tests and t-test should be reported in a consistent format throughout the manuscript (e.g., APA format).
**Response:** We have changed the reporting of the results to match the APA formatting style.

8) In its current form, Appendix B lacks structure and contains several unclear phrases, e.g., "Your local environment - When do you think the climate crisis will start to affect the following?" – here, 'your local environment' should be placed after the question.

**Response:** We have restructured the supplementary material and cut back items that are not relevant to the article.